# Dietary fiber content in clinical ketogenic diets modifies the gut microbiome and seizure resistance in mice

Ezgi Özcan [1,3] ✉, Kristie B. Yu[1], Lyna Dinh[1], Gregory R. Lum[1], Katie Lau[1], Jessie Hsu [1], Mariana Arino[1], Jorge Paramo[2], Arlene Lopez-Romero [2] & Elaine Y. Hsiao [1,2] ✉

The gut microbiome modulates the anti-seizure effects of the ketogenic diet, but how specific dietary formulations differentially modify the gut microbiome in ways that impact seizure outcome is poorly understood. We find that medical ketogenic infant formulas vary in macronutrient ratio, fat source, and fiber content and differentially promote resistance to 6-Hz seizures in mice. Dietary fiber, rather than fat ratio or source, drives substantial metagenomic shifts in a model human infant microbial community. Addition of fiber to a fiber-deficient ketogenic formula restores seizure resistance, and supplementing protective formulas with excess fiber potentiates seizure resistance. By screening 13 fiber sources and types, we identify metagenomic responses in the model community that correspond with increased seizure resistance. Supplementing with seizure-protective fibers enriches microbial genes related to queuosine biosynthesis and $preQ_0$ biosynthesis and decreases genes related to sucrose degradation and TCA cycle, which are also seen in seizure-protected mice that are fed fiber-containing ketogenic formulas. This study reveals that different formulations of ketogenic diets, and dietary fiber content in particular, differentially impact seizure outcome in mice, likely by modifying the gut microbiome. Understanding interactions between diet, microbiome, and host susceptibility to seizures could inform novel microbiome-guided approaches to treat refractory epilepsy.

The low-carbohydrate, high-fat ketogenic diet (KD) is used to treat epilepsy in children who do not respond positively to existing anti-seizure medications. While it is well integrated into the healthcare system, KD therapies have variable effectiveness in reducing seizures, ranging from 45 to 85% in infants and children that exhibit high compliance[1–5] and with substantially lower rates in adults[6,7]. Recent reports highlight a key role for the gut microbiome in mediating the effects of the KD on various host physiologies, including glucose and lipid metabolism[8], immune function[9,10], brain activity[11], and behavior[12,13]. The KD alters the gut microbiome across several human and animal epilepsy studies[14–19], and relationships are seen between the gut microbiome and seizure resistance in various rodent epilepsy models[12,20–23]. Findings from the field are converging upon the notion that variation in the gut microbiome may contribute to variability in patient responsiveness to the KD, and that microbiome-targeted interventions could be used to promote the efficacy of the KD in treating refractory epilepsy.

While existing studies of the microbiome and KD have focused predominantly on the classic KD, many variations of the KD with

[1]Department of Integrative Biology & Physiology, University of California, Los Angeles, CA, USA. [2]UCLA Goodman-Luskin Microbiome Center, Vatche and Tamar Manoukian Division of Digestive Diseases, David Geffen School of Medicine, Los Angeles, CA, USA. [3]Present address: School of Nutrition and Food Sciences, Louisiana State University Agricultural Center, Baton Rouge, LA, USA. ✉e-mail: eozcan@agcenter.lsu.edu; ehsiao@g.ucla.edu

different macronutrient ratios and types are used clinically to treat epilepsy, depending on factors such as the age of the patient, seizure type, and tolerability of the dietary regimen[24–27]. For example, the KD is commonly administered as a 4:1 or 3:1 fat-to-carbohydrate and protein ratio, depending on patient tolerance. The medium chain triglyceride (MCT) diet, often derived from MCT-rich coconut oil, is thought to promote enhanced ketone production while being less restrictive than the classic KD. The Modified Atkins Diet (MAD), which does not require strict weighing of food or fluids, and Low Glycemic Index Treatment (LGIT), which focuses on carbohydrates with low glycemic index rather than removal, are additional less restrictive variations of the KD that are frequently used in older children and adults.

While only a few small human studies have compared different KD variants for their seizure reduction, citing no significant differences[25,26,28,29], other research suggests that differences in dietary formulation may impact host responses to the KD. In a retrospective open-label trial of patients with drug-resistant epilepsy, transitioning to a polyunsaturated fatty acid (PUFA)-based KD enhanced seizure control in individuals who responded poorly to the classic KD[30]. Moreover, differences in dietary formulation can have substantial impacts on microbiome-dependent host phenotypes – KDs with different fat ratio and/or source resulted in differential influences of the microbiome on host glucose and lipid metabolism, as well as immune function[8,9]. In addition, supplementation with dietary fiber, a key energy source for gut bacteria that modulates myriad host metabolic, immune, and neural functions, is incorporated into some clinical KD regimens to ease gastrointestinal symptoms[31], but whether it alters seizure response is unclear[32]. Overall, increasing research indicating that the gut microbiome modifies seizure susceptibility and the anti-seizure effects of the KD raises the important question of how variations in the formulation of medical KDs differentially shape the microbiome in ways that impact seizure outcome.

In this study, we tested effects of three clinically prescribed KD infant formulas on the mouse gut microbiome and resistance to 6-Hz psychomotor seizures, as a benchmark model of refractory epilepsy[33]. To determine which dietary variables serve as key drivers of microbiome response, we established a model human infant microbial consortium and assessed effects of fat ratio, fat source, and carbohydrate source on shaping its functional potential. We further screened 13 fiber sources and types for their differential impacts on the model infant microbial community and tested top candidates for their ability to restore and/or potentiate seizure-protective effects of clinical KD infant formulas. Results from this study reveal key diet-microbiome interactions that promote the seizure-protective effects of medical KDs.

## Results

### Different clinical KD infant formulas elicit differential seizure responses in mice

Mechanistic studies of the KD on seizure resistance often rely on commercial KD chows that are formulated for lab animals and not directly relevant to medical KD therapies used for human epilepsy. At the same time, clinical KD regimens vary widely in nutritional content and are often tailored to the particular individual's needs and tolerability, making it difficult to identify standard regimens. To examine how clinically relevant formulations of the KD elicit differential effects on seizure outcome, we focused on 3 commonly prescribed commercial KD infant formulas – KD4:1, KD3:1, and MCT2.5:1 -- due to their reproducible composition, direct clinical relevance, frequent prescription, and importance for infants and young children as especially vulnerable subsets of refractory epilepsy patients for which improved interventions are needed. Compared to a standard infant formula as a control diet (CD), the 3 KD infant formulas all exhibit high-fat content relative to carbohydrate and protein, but they display nuanced

differences in formulation (Fig. 1a, Supplementary Data 1). In addition to differences in fat ratio, fat source varies between the formulations, where KD4:1 contains soy lecithin but lacks coconut oil (MCT source) and linoleic acid, KD3:1 contains linoleic acid but lacks soy lecithin and coconut oil, and MCT2.5:1 contains coconut oil but lacks soy lecithin and linoleic acid. There are also differences in carbohydrate content, where both KD4:1 and MCT2.5:1 contain corn syrup solids, high amylose corn starch, chicory root inulin, gum arabic, cellulose, fructooligosaccharides (FOS), soy fiber, and maltodextrin, whereas KD3:1 contains only lactose and corn syrup solids, with none of the dietary fibers. The CD contains lactose and less than 2% dietary fiber comprised of galactooligosaccharides, which differs from the types of fibers included in KD4:1 and MCT2.5:1.

To determine how different KD formulations impact seizure susceptibility, we fed cohorts of conventional 4-week-old mice the KD4:1, KD3:1, MCT2.5:1, or CD formula as liquid diet for 1 week, and then tested for susceptibility to 6-Hz psychomotor seizures (Fig. 1b). Juvenile mice were selected to mimic the typical use of the KD to treat pediatric epilepsy, to align the timing of mouse brain development to early brain development in humans[34], and to preclude effects of pre-weaning treatment, where effects of the diets on maternal behavior and physiology would confound their direct effects on offspring. 1 week of feeding was selected based on our prior longitudinal characterization, which indicated that KD chow shifts the gut microbiome and confers seizure protection by day 4 of treatment in mice[12]. Finally, the 6-Hz seizure assay was selected as a benchmark model of refractory epilepsy that is used to screen for new anti-seizure medications[33] and involves low-frequency corneal stimulation to induce complex partial seizures related to human temporal lobe epilepsy[35]. KD chow protects against 6-Hz seizures, as indicated by increases in current intensity required to elicit a seizure in 50% of the subjects tested (CC50, seizure threshold)[12,36,37].

As seen previously for KD chows[12,36,37], we observed that feeding mice clinical KD4:1 infant formula increased seizure thresholds compared to controls fed a CD infant formula (Fig. 1c). MCT2.5:1 also increased seizure thresholds albeit to a lesser degree than KD4:1, which may be due to its comparatively lower fat ratio or different fat source. In contrast, however, KD3:1 infant formula yielded decreased seizure thresholds compared to all other groups, including CD-fed controls, suggesting that the KD3:1 formulation increases susceptibility to 6-Hz seizures in mice. There was no correlation of seizure threshold with average calories consumed for the different KDs or with degree of ketosis as assessed by serum levels of beta-hydroxybutyrate (Supplementary Fig. 1a, b). To further assess whether the differences in seizure outcome may be confounded by nuances of providing the diet in liquid form, such as differences in density or leakage from the bottle, we repeated the experiment by providing the infant formula diets in solid form following dehydration. Consistent with our previous observation, solid KD4:1 and MCT2.5:1 increased seizure threshold relative to controls fed solid CD, whereas solid KD3:1 decreased resistance to 6-Hz seizures, with no correlation with total diet consumed (Supplementary Fig. 1c, d). These data indicate that variations in clinical KD formulations differentially modify host resistance versus susceptibility to 6-Hz seizures in mice.

### Clinical KD infant formulas differentially alter the mouse gut microbiome

Classic KD-induced changes in the mouse and human microbiome are necessary and/or sufficient to confer resistance to 6-Hz seizures in mice[12,20]. To determine how the different clinical KD infant formulas impact the gut microbiome, we performed metagenomic sequencing of fecal microbiota from mice fed KD4:1, KD3:1, MCT2.5:1, or CD for 1 week. In contrast to results from KD vs. standard chow[12,38], KD4:1 and MCT2.5:1 significantly increased α-diversity of the microbiome, as indicated by elevated Shannon's diversity index, when compared to

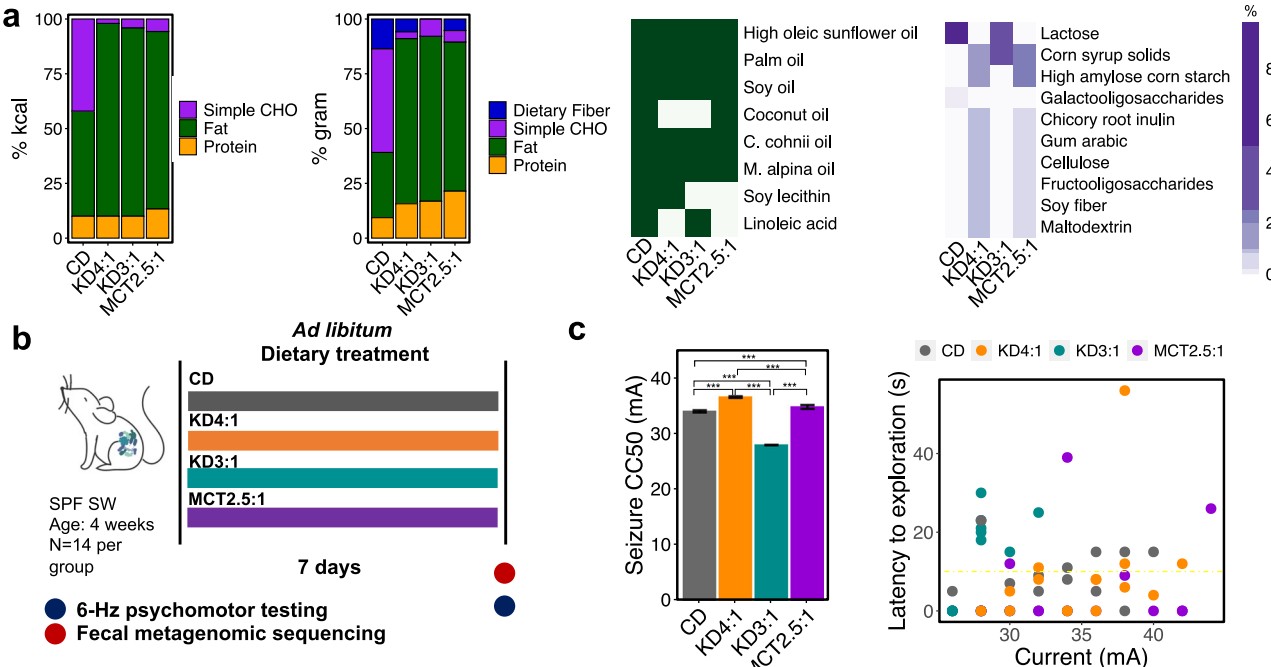

**Fig. 1 | Different formulations of medical ketogenic diets (KD) elicit differential responses to 6-Hz seizures in mice. a** Macronutrient composition without fiber (for determining KD fat ratio), macronutrient composition with fiber, absence/presence of fat sources, and percent carbohydrate composition for the commercial KD infant formulas KD4:1, KD3:1, and MCT2.5:1, relative to standard infant formula as control diet (CD). **b** Experimental design: 4 week old conventional (specific pathogen free, SPF) Swiss Webster (SW) mice ($n = 14$ mice/group) were fed each medical KD or CD as liquid diets for 7 days. **c** 6-Hz seizure threshold (left) and latency to exploration (right) for mice fed KDs or CD as liquid diet (left, one-way ANOVA with Bonferroni, $n = 14$ mice/group, ***$p < 0.001$). Data are presented as mean ± SEM. Yellow line at $y = 10$ s represents threshold for scoring seizures. Data are provided as a Source Data file.

CD controls (Fig. 2a). However, there was no significant effect of KD3:1 on Shannon diversity levels, despite comparable increases across all KD formula groups in species richness of the fecal microbiota. This suggests that the main driver of α-diversity differences between the KD groups is differential alteration in species evenness−indeed, KD3:1 yielded fecal microbiota with significantly reduced Pielou's evenness compared to KD4:1 and MCT2.5:1 groups. β-diversity analysis of the gut microbiota based on Aitchison distance showed that KD3:1 clustered distinctly from the CD controls and KD4:1 along PCoA2 and from MCT2.5:1 along PCoA1 (PERMANOVA, $p = 0.001$, $R^2 = 0.31$, Fig. 2b), whereas Bray-Curtis dissimilarity and weighted Unifrac distances showed that KD samples clustered distinctly from CD controls along PCoA1, with KD4:1 and MCT2.5:1 samples showing further separation from CD than KD3:1 samples (PERMANOVA, $p = 0.001$, $R^2 = 0.6$, Fig. 2b). In particular, all KD groups exhibited significantly decreased relative abundances of *Actinobacteria* and increased *Bacteroidetes* and unclassified bacteria compared to CD controls (Supplementary Fig. 2a, Supplementary Data 7). However, only KD4:1 and MCT2.5:1 shared statistically significant decreases in *Erysipelotrichia* and increases in *Streptococcaceae*, *Coriobacteriia*, and *Deferribacteres*, whereas KD3:1 exhibited no significant changes in these taxa compared to CD (Supplementary Fig. 2a–c). Rather, KD3:1 showed significantly increased relative abundance of *Proteobacteria*, *Escherichia coli*, *Enterococcus faecalis*, and *Mammaliicoccus sciuri* compared to CD, KD4:1, and/or MCT2.5:1 (Supplementary Fig. 2a–c).

The seizure-susceptible KD3:1 group also exhibited decreased representation of the top 10 most abundant metagenomic superclass pathways (Fig. 2c), suggesting that the KD3:1 limits the presence of microbial taxa associated with prevalent functions and/or enriches the representation of previously rare metagenomic pathways. Among the top 10, the relative abundance of superclass pathways related to amino acid, carbohydrate, and nucleoside and nucleotide biosyntheses were significantly lower in KD3:1 relative to MCT2.5:1, CD, and/or KD4:1

groups. In contrast, superclass pathways related to carboxylic acid, fatty acid and lipid, and secondary metabolite degradation were significantly elevated in KD3:1 compared to other groups. When considering specific alterations at the more resolved pathway level, all three KDs shared subsets of metagenomic changes compared to CD controls, where KD4:1 and MCT2.5:1 shared greater overlap than with KD3.1 (Fig. 2d). Namely, KD4:1 and MCT2.5:1 (but not KD3:1) similarly induced significant metagenomic increases in select pathways related to carbohydrate biosynthesis (UDP-N-acetyl-D-galactosamine II and UDP-N-acetyl-D-glucosamine biosynthesis II), carboxylic acid degradation (biotin-dependent malonate degradation), and cofactor, carrier, and vitamin biosynthesis (biotin biosynthesis), and decreases in select pathways related to carbohydrate degradation (hexitol and galactitol degradation, sucrose, lactose, galactose degradation, and Entner-Doudoroff pathway), amino acid biosynthesis (L-lysine and L-alanine biosynthesis), carbohydrate biosynthesis (UDP-N-acetyl-D-glucosamine biosynthesis I and UDP-glucose-derived-O-antigen building blocks biosynthesis), and pentose phosphate pathway compared to CD controls (Fig. 2e). KD3:1 displayed the most differentially abundant metagenomic pathways compared to CD, which were distinct from those seen in the other KD groups (Supplementary Fig. 2d). The majority of differentially abundant pathways that were elevated by KD3:1 related to amide, amidine, amine, and polyamine degradation, fatty acid and lipid biosynthesis, carboxylic acid degradation, and fermentation (Supplementary Fig. 2d). In particular, pathways for phospholipid remodeling, lactate fermentation, and biosynthesis of octanoyl and myristate, and degradation of erythronate, threonate, galactitol, and allantoin were all significantly increased by KD3:1, decreased by KD4:1 and MCT2.5:1 (Supplementary Fig. 2d), and associated with low dietary fiber content (Supplementary Fig. 2e). The only pathway decreased by KD3:1, but elevated by KD4:1 and MCT2.5:1, was L-glutamate and L-glutamine biosynthesis (Supplementary Fig. 2d), which was further positively associated with dietary fiber

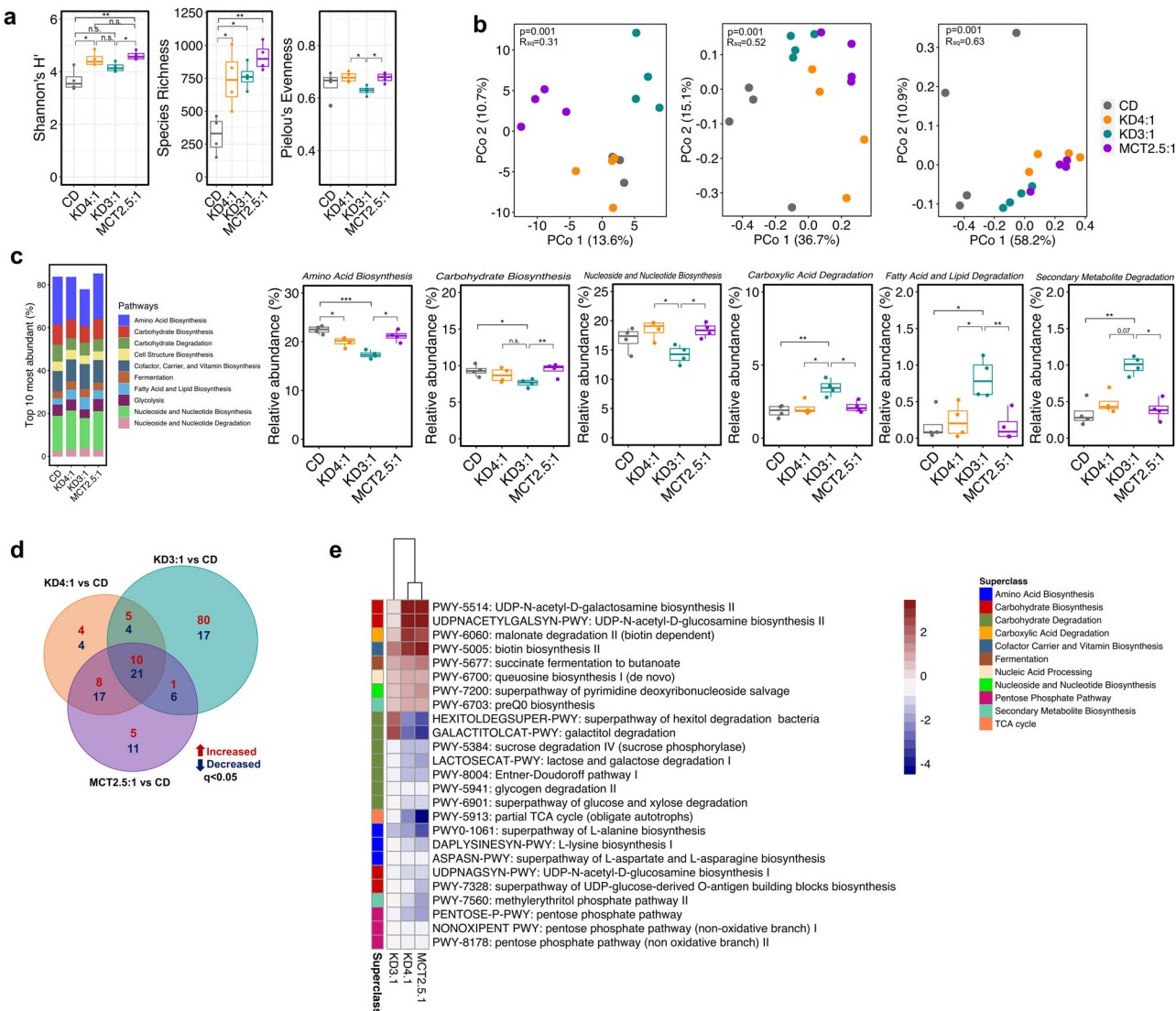

**Fig. 2 | Medical KDs induce differential alterations in the gut microbiome that associate with resistance vs. susceptibility to 6-Hz seizures. a** Alpha diversity from fecal metagenomic sequencing data after treatment with KDs or CD (Kruskal-Wallis with Dunn's test: *$p < 0.05$, **$p < 0.01$, n.s. not statistically significant; $n = 4$ cages/group. Data are presented as box-and-whisker plots with median and first and third quartiles). **b** Principal coordinates analysis (PCoA) of Aitchison distance (left), Bray-Curtis dissimilarity (middle), and weighted UniFrac distance (right) based on fecal metagenomic sequencing data after dietary treatment (PERMANOVA, one-sided, $n = 4$ cages/group). **c** Top 10 most abundant metagenomic superclass pathways (left). Differentially abundant pathways that are significantly altered in seizure susceptible group KD3:1 and/or shared between seizure protected groups KD4:1

and MCT2.5:1 (right, Kruskal-Wallis with Dunn's test: *$p < 0.05$, **$p < 0.01$, ***$p < 0.001$; $n = 4$ cages/group. Data are presented as box-and-whisker plots with median and first and third quartiles). **d** Venn diagram of differential metagenomic pathways ($q < 0.05$) for each KD relative to CD. (MaAsLin2, two-sided, General Linear Model (GLM); $n = 4$ cages/group). **e** Heatmap of differential metagenomic pathways ($q < 0.05$) that are shared between seizure-protected groups KD4:1 and MCT2.5:1 and not significant in seizure-susceptible group KD3:1 (two-sided, GLM statistical test, $n = 4$/condition). Color scale represents the coefficient value from MaAsLin2 output within the feature table. All taxonomic relative abundances at SGB-level and MetaCyc pathway abundances used to generate the data are provided in Supplementary Data 7 and 8. Data are provided as a Source Data file.

(Supplementary Fig. 2e). Taken together, these results indicate that resistance vs. susceptibility to 6-Hz seizures in response to different KD infant formulas is associated with differential alterations in the composition and functional potential of the gut microbiome.

### Fiber content in the KD drives microbial alterations and promotes seizure resistance

The gut microbiome is shaped by changes in host diet and can be responsive to the presence, abundance, and sources of dietary macronutrients[39]. To gain insight into how different clinical KD formulas differentially alter the gut microbiome, we screened various dietary parameters for their effects on a model human infant microbial community. 9 bacterial strains were selected based on their prevalence

and relative abundances across multiple large studies of the infant gut microbiome[40,41] (Supplementary Fig. 3a, Supplementary Data 3). All community members were confirmed to grow stably together in a rich complex medium[42] as a positive control (Supplementary Fig. 3b). To test the effects of KD fat ratio, the model infant gut microbial community was cultured in synthetic KD media prepared in ratios from KD4:1 to KD1.5:1 (Supplementary Fig. 3c, Supplementary Data 6). There were no statistically significant differences in taxonomic response to the KDs with different fat ratio (PERMANOVA, $p = 0.13$, $R^2 = 0.14$, Supplementary Fig. 3d). To examine effects of KD fat source, the model infant gut microbial community was cultured in synthetic media representing KD4:1, KD3:1, or MCT2.5:1, each using sunflower oil (6% saturated fat), soy lecithin (23% saturated fat and dominant in KD4:1

infant formula), or palm oil (50% saturated fat), as fat sources with different levels of saturation (Supplementary Fig. 3e). The media prepared with soy lecithin increased the absolute abundance of *B. infantis*, *B. fragilis*, and *C. perfringens*, resulting in distinct separation along PCoA1 from the sunflower and palm oil groups (PERMANOVA, $p < 0.05$; Supplementary Fig. 3f). This may be due to the presence of free sugars (8%) in the commercial soy lecithin and/or the emulsifying properties of soy lecithin, compared to the other fat sources[43]. There were no statistically significant differences between the sunflower and palm oil groups across all media conditions (Supplementary Fig. 3f), suggesting that the differential effects of soy lecithin are driven by fat source rather than saturation level.

To test effects of additional fat sources, KD-based media were also prepared with addition of MCT, dominant in MCT2:5:1 infant formula, or linoleic acid, dominant in KD3:1 infant formula (Supplementary Fig. 3g). Addition of MCT increased the absolute abundance of *B. breve*, *B. infantis*, and *B. longum* compared to corresponding controls, resulting in notable shifts in diversity when added to KD4:1 and KD3:1 media (PERMANOVA $p = 0.05$, $R^2 = 0.33$; $p = 0.017$, $R^2 = 0.32$), but not KD2.5:1 media (PERMANOVA $p = 0.55$, $R^2 = 0.04$) (Supplementary Fig. 3h). In contrast, addition of linoleic acid decreased the absolute abundance of *B. infantis* and *B. vulgatus*, which resulted in statistically significant shifts across PCoA1 relative to all media groups (Supplementary Fig. 3h). This raises the question of whether differential effects of linoleic acid on the microbiome could contribute to the failure of KD3:1 infant formula to protect against 6-Hz seizures (Fig. 1c, Supplementary Fig. 1c).

Finally, to evaluate the effects of carbohydrate type, the model infant gut microbial community was cultured in synthetic media representing KD4:1, KD3:1, and MCT2.5:1 and containing either lactose or a fiber mix, comprised of equal amounts of FOS, inulin, cellulose, and gum arabic, as the fiber sources that distinguish KD4:1 and MCT2.5:1 infant formula from KD3:1 and CD formulas (Fig. 3a). The presence of dietary fiber led to substantial shifts in the model infant gut microbial community across all media conditions, with particular enrichment of *B. fragilis* and decreases in *B. breve* and *B. infantis* (Supplementary Fig. 3i). PCoA analysis of synthetic metagenomic data (Supplementary Data 9) assembled from quantitative taxonomic profiles showed notable clustering of fiber mix groups away from lactose controls (PERMANOVA, $p = 0.02$ (KD4:1), $p = 0.08$ (KD3:1), $p = 0.001$ (MCT2.5:1), Fig. 3b), with greater discrimination than seen with alterations in fat ratio or source (Supplementary Fig. 3d–h). In particular, fiber mix yielded statistically significant decreases in several pathways related to amino acid biosynthesis, nucleotide and nucleoside biosynthesis, and carbohydrate degradation, among many others (Fig. 3c and Supplementary Fig. 4). Among the 110 metagenomic pathways that were significantly altered by in vitro culture of the model infant microbial community with fiber mix compared to lactose (Supplementary Fig. 4), 15 pathways (13.6%) were similarly significantly altered in the fecal microbiome of mice fed the fiber-containing KD4:1 and MCT2.5:1, as compared to lactose-containing CD controls (Fig. 3c). Specifically, queuosine biosynthesis and its intermediate preQ$_0$ biosynthesis were significantly enriched by fiber in the in vitro system and by fiber-containing KDs in the mouse. Similarly, fiber-induced decreases in pentose phosphate pathways, pathways related carbohydrate degradation (sucrose, glucose, xylose, and glycogen degradation), carbohydrate biosynthesis (UDP-N-acetyl-D-glucosamine biosynthesis and UDP-glucose derived O-antigen building blocks biosynthesis), amino acid biosynthesis (L-alanine, L-lysine and L-aspartate and L-asparagine biosynthesis), partial TCA cycle, and methylerythritol phosphate pathway were also shared with mouse metagenomes of KD4:1 and MCT2.5:1 groups (Figs. 3c, 2e). The results suggest that dietary fiber, more so than fat ratio or source, exerts a strong influence on community structure and functional potential of a model infant gut microbial community. Select alterations are consistent with those seen

in the mouse microbiome in response to host consumption of fiber-containing clinical KD infant formulas (KD4:1 and MCT2.5:1), which confer resistance to 6-Hz seizures. The results suggest that these particular metagenomic signatures may be related to seizure resistance.

To test whether dietary fiber content has a causal impact on resistance to 6-Hz seizures, we supplemented the fiber mix into the KD3:1 infant formula to match reported fiber levels in KD4:1 infant formula, and tested mice for seizure susceptibility at 7 days after dietary treatment (Fig. 3d). As previously demonstrated, mice fed liquid KD3:1 exhibited decreased seizure threshold compared to CD controls (Fig. 3e). Notably, addition of fiber to the KD3:1 elevated seizure thresholds to levels that exceeded those seen in CD controls. We further repeated the fiber supplementation using the solid diet paradigm, where the same infant formulas were dehydrated and administered as chow instead of liquid diet. As seen in liquid form, supplementation with fiber mix significantly increased seizure threshold of mice fed KD3:1, with no significant differences in diet consumption (Supplementary Fig. 5a, b). Similar outcomes were seen in both male and female mice ($p < 0.001$), indicating that the ability of fiber supplementation to promote seizure resistance in mice fed KD3:1 formula is not sex dependent (Supplementary Fig. 5c, d). We further investigated whether adding excess fiber to CD formula could enhance seizure protection. Fiber supplementation to CD-fed mice resulted in only a very minimal increase (CC50_CD = 48.19, CC50_CD + Fiber = 48.67, $p < 0.01$) (Supplementary Fig. 6), suggesting that fiber supplementation is more effective in the context of KD consumption. These data demonstrate that addition of fiber to the low fiber KD3:1 infant formula restores its antiseizure effects toward levels seen with fiber-containing KD4:1 and MCT2.5:1.

To determine whether dietary fiber supplementation can potentiate KD-induced seizure protection, we supplemented the fiber-containing KD4:1 infant formula, which yielded the highest seizure thresholds of all KD variants (Fig. 1), with the dietary fiber mix that is already existing in the formula and tested mice for resistance to 6-Hz seizures after 7 days of feeding with the liquid diet (Fig. 4a). The additional fiber added to KD4:1 formula increased fiber content from 5.3% to ~10.3%. Dietary fiber supplementation significantly increased seizure thresholds to levels that exceeded those seen with KD4:1 alone (Fig. 4b). There were no significant differences between groups in dietary consumption (Supplementary Fig. 7a). The ability of fiber supplementation to further promote the anti-seizure effects of KD4:1 was similarly seen when administered as solid diet, instead of liquid diet, also with no significant differences in food consumption (Supplementary Fig. 7b, c). Short-chain fatty acids (SCFAs) are primary end products of gut microbial fiber fermentation in the colon and have been shown to impact host brain activity and behavior[44]. To further ask whether fiber supplementation promotes seizure resistance via SCFAs, we supplemented KD4:1 infant formula with the SCFAs acetate, butyrate, and propionate, at concentrations predicted to match those produced by fermentation of the dietary fiber mix. In both liquid and solid form, SCFA supplementation failed to phenocopy effects of dietary fiber supplementation and instead yielded mice with modest reductions in resistance to 6-Hz seizures, as compared to controls supplemented with vehicle solution (Supplementary Fig. 8a, b). Notably, we detected elevations in only serum acetate concentrations when SCFAs were supplemented in solid form in the paste diet (Supplementary Fig. 8c, d), which may be attributable to the rapid absorption, utilization, and distribution of exogenously delivered SCFAs[45–47]. Consistent with this, we observed no overt differences in SCFA levels in fiber-supplemented mice fed KD (Supplementary Fig. 9), which may be due to variations in the timing of food intake and fiber metabolism relative to sample collection[45]. To explore additional non-SCFA metabolites that may mediate effects of fiber mix, we profiled an additional 93 cecal metabolites from mice fed KD4:1, with or without fiber supplementation (Supplementary Data 11). Mice fed fiber-supplemented

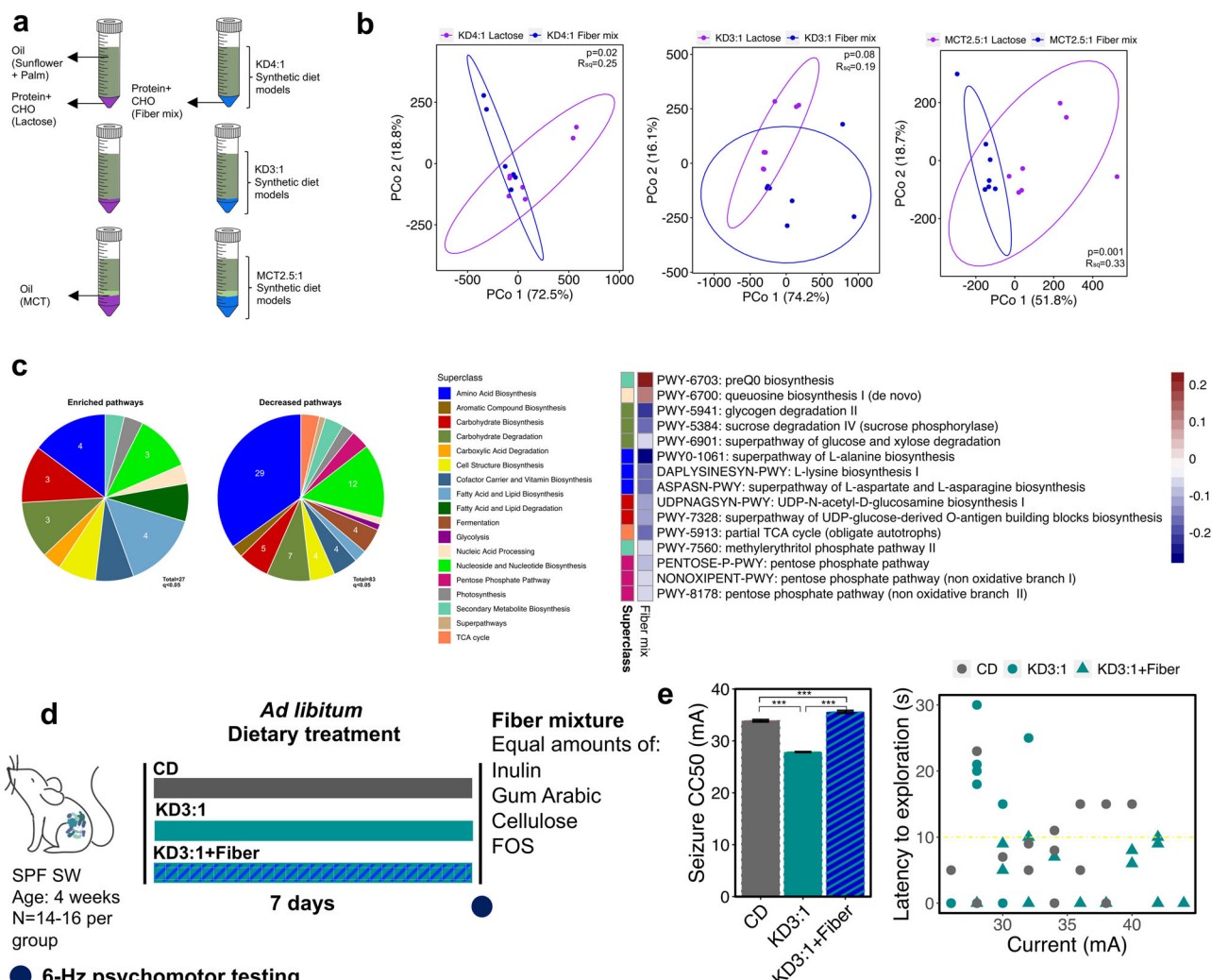

**Fig. 3 | Addition of dietary fiber to KDs enriches metagenomic features associated with seizure protection in a model human infant gut community and restores resistance to 6-Hz seizures in mice. a** Experimental design: Fiber mix containing inulin, gum arabic, cellulose, and fructooligosaccharide (FOS), or lactose as a non-fiber carbohydrate control, was added to KD-based synthetic culture media for anaerobic culture of a model human infant microbial community. **b** Principal coordinates plots of metagenomic pathway abundance data for human infant microbes grown in KD-based media containing fiber mix versus lactose (PERMANOVA, one-sided, $n = 7$/condition). **c** Venn diagram of differential metagenomic pathways (q < 0.05) shared across all fiber-containing KD media groups relative to corresponding lactose-containing media groups as controls (left). 15 fiber-induced differential metagenomic pathways (q < 0.05) that are similarly seen in seizure-protective mice fed KD4:1 or MCT2.5:1 (right, two-sided, General Linear Model, $n = 7$/condition). **d** Experimental design: 4-week-old conventional (specific pathogen free, SPF) Swiss Webster (SW) mice ($n = 14$–16 mice/group) were fed KD3:1 supplemented with fiber mix, KD3:1 alone, or CD as liquid diets for 7 days. **e** 6-Hz seizure threshold (left) and latency to exploration (right) for mice fed KD3:1+fiber mix, KD3:1, or CD as liquid diet (left, one-way ANOVA with Bonferroni: ***$p < 0.001$; $n = 14$ mice/group). Data are presented as mean ± SEM. Yellow line at y = 10 s represents threshold for scoring seizures. Pathway abundances used to generate the data are provided in Supplementary Data 9. Data are provided as a Source Data file.

KD4:1 had significantly increased cecal levels of 2-hydroxyglutarate and decreased levels of L-aspartate, hypoxanthine, xanthine, inosine, and uridine (Supplementary Fig. 10), which aligns with metagenomic alterations in genes related to TCA cycle and L-aspartate biosynthesis (Fig. 3c). These results additionally implicate alterations in adenosine and uracil metabolism. Overall, these data indicate that dietary fiber supplementation both restores the anti-seizure effects of the low fiber KD3:1 and further potentiates the anti-seizure effects of the fiber-containing KD4:1, through mechanisms that are not recapitulated by oral SCFA supplementation.

### Different fiber types and sources elicit differential microbial alterations and seizure outcomes

Dietary fibers are fermented by select gut bacteria and shape the composition and activity of the gut microbiome[48]. To gain insight into whether particular fiber types or sources interact with KD4:1 to differentially alter the infant gut microbiome, we screened 13 different fiber conditions, comprised of commercially available fiber products or purified fiber types, for their additional effects on the model infant microbial community when grown directly in KD4:1 infant formula (rather than in a diet-based synthetic culture medium, as in prior experiments) (Fig. 4c). Taxonomic profiles showed that 8 out of the 13 fiber conditions significantly increased the absolute abundance of *B. fragilis*, and 11 fiber conditions significantly decreased *B. breve* (Supplementary Fig. 11), both of which align with previous in vitro results from fiber supplementation into synthetic media (Supplementary Fig. 3i). 7 of the 13 fiber conditions yielded reductions in *E. coli*, which parallel the increases in *E. coli* observed with mouse consumption of fiber-deficient KD3:1 (Supplementary Fig. 2c). We next generated synthetic metagenomic profiles for the 13 fiber supplementation conditions (Supplementary Data 10) and filtered results to prioritize the 15 protective features that were

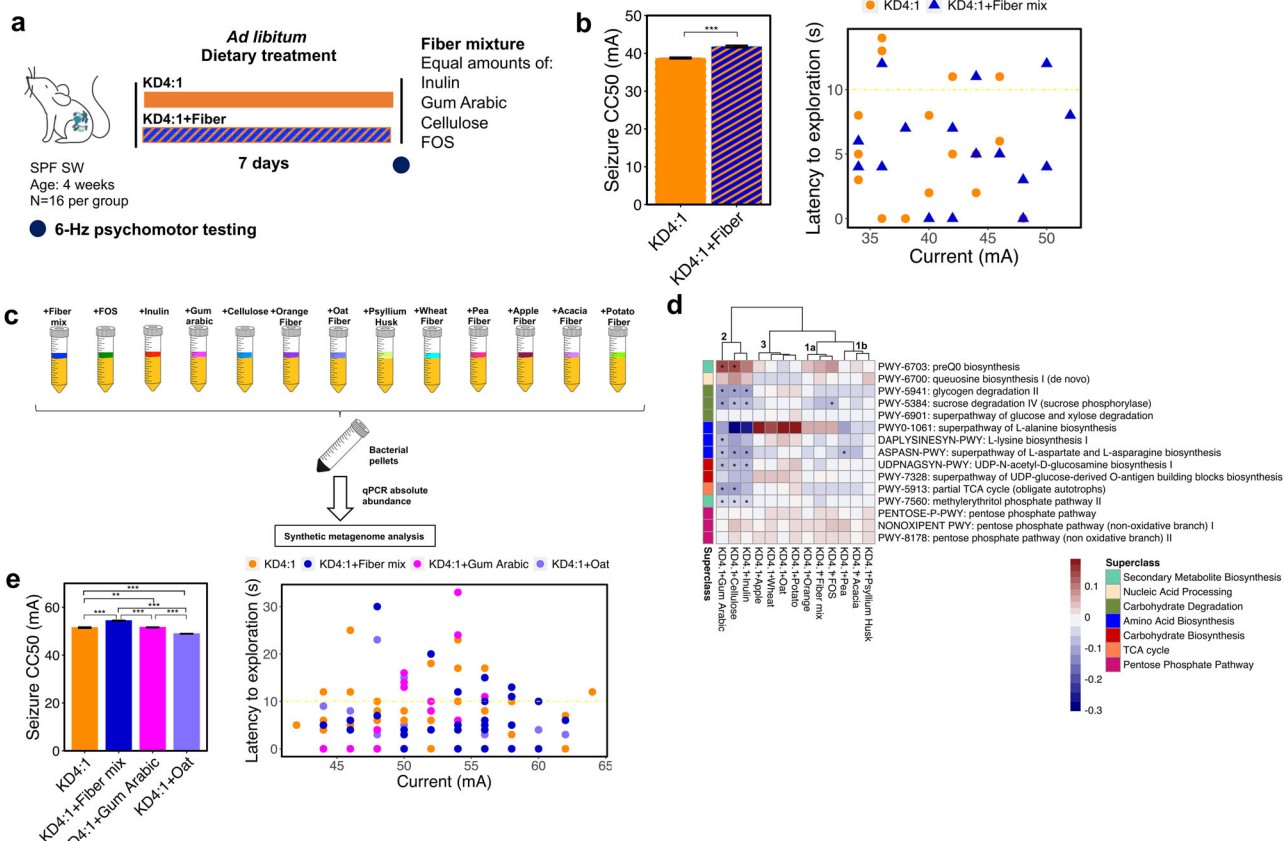

**Fig. 4 | Addition of excess dietary fiber to fiber-containing KD4:1 further potentiates seizure resistance. a** Experimental design: 4-week-old conventional (specific pathogen free, SPF) Swiss Webster (SW) mice ($n = 16$ mice/group) were fed KD4:1 supplemented with fiber mix or KD4:1 alone as liquid diets for 7 days. **b** 6-Hz seizure threshold (left) and latency to exploration (right) for mice fed KD4:1 and KD4:1+fiber mix as liquid diet (left, two-sided Welch's $t$-test: ***$p < 0.001$; $n = 16$ mice/group). Yellow line at y = 10 s represents threshold for scoring seizures. **c** Experimental design: 13 dietary fiber sources and types were supplemented to KD4:1 infant formula for anaerobic culture of a model human infant gut microbial community. **d** Heatmap of 15 fiber-induced differential metagenomic pathways ($q < 0.05$) that were similarly seen in seizure-protected mice fed KD4:1 or MCT2.5:1 (right). Groupings were denoted on top of the dendrogram (two-sided, General

Linear Model statistical test, $n = 8–10$/condition, *$q < 0.05$ for fiber source/type relative to KD4:1 as a control). Color scale represents the coefficient values from MaAslin2 output within the features table. **e** 6-Hz seizure threshold (left) and latency to exploration (right) for mice fed KD4:1 supplemented with dietary fiber mix (Group 1), gum arabic (Group 2), or oat fiber (Group 3), or KD4:1 alone as paste diet (left, one-way ANOVA with Bonferroni: **$p < 0.01$, ***$p < 0.001$; $n = 16$ mice per gum arabic and oat, $n = 48$ per KD4:1 and $n = 32$ per KD4:1+fiber mix group where multiple independent experiments were combined). Data are presented as mean ± SEM. Yellow line at y = 10 s represents threshold for scoring seizures. Pathway abundances used to generate the data are provided in Supplementary Data 10. Data are provided as a Source Data file.

shared between mouse consumption of the KD4:1 and MCT2.5:1 (Fig. 2e) and model human infant microbial community responses to fiber in synthetic diet-based media (Fig. 3c, Supplementary Fig. 4). The results revealed 4 subgroupings of model infant microbial responses to the 13 different fibers in KD4:1 infant formula (Fig. 4d). Group 1a consisted of fiber mix, FOS, and orange fiber and was characterized by increases in genes related to preQ0 biosynthesis and L-alanine biosynthesis, with reductions in sucrose degradation and partial TCA cycle (Fig. 4d). Group 1b consisting of pea, acacia, and psyllium husk fibers, clustered together with Group 1a and exhibited a similar general pattern of metagenomic features but with reductions in L-alanine biosynthesis and less substantial shifts in preQ0 biosynthesis and sucrose degradation (Fig. 4d). Group 2 consisted of inulin, cellulose, and gum arabic, which was characterized by significant decreases in genes related to 5–7 pathways (glycogen and sucrose degradation, L-alanine, L-lysine, L-aspartate, L-asparagine, and UDF-N-acetyl-D-glucosamine biosynthesis, partial TCA cycle, and methylerythritol phosphate pathway) and significant increases in preQ0 biosynthesis genes (Fig. 4d). Group 3, consisting of oat, potato, wheat, and apple fibers, was characterized by notable increases in representation of L-alanine biosynthesis and UDP-

glucose-derived O-antigen building blocks biosynthesis, with decreases in queuosine biosynthesis (Fig. 4d).

Based on these patterns of microbial representation for key metagenomic features conserved in mice fed fiber-containing KDs and infant microbial communities cultured with fiber-supplemented media, we selected one representative fiber condition per primary grouping (Group 1: fiber mix, Group 2: gum arabic, Group 3: oat fiber) to test for causal effects on seizure resistance. We supplemented representative fibers from each group into KD4:1 infant formula to raise fiber content from 5.3% to ~10.3%, and tested mice for resistance to 6-Hz seizures at the 7th day after feeding in paste form. As previously observed in liquid and solid diet form (Fig. 4b, Supplementary Fig. 7b), supplementation of KD4:1 paste with fiber mix significantly increased resistance to 6-Hz seizures (Fig. 4e). In contrast, supplementation with gum arabic (Group 2) had no overt effects on seizure threshold compared KD4:1 controls (Fig. 4e). In addition, supplementation with oat fiber (Group 3) had a detrimental effect, significantly decreasing seizure thresholds compared to KD4:1 controls and all other fiber conditions (Fig. 4e). Overall, these data reveal that the ability of fiber supplementation to potentiate the seizure protective effects of KD4:1 infant formula is specific to particular sources and

types of fibers that alter key metagenomic features of the gut microbiome.

## Discussion

Findings from this study demonstrate that different clinical KD infant formulas have varying effects on seizure resistance in mice, likely due to differences in how specific dietary components affect the function of the gut microbiome. We find that fiber-containing commercial infant formulas KD4:1 and MCT2.5:1 promote resistance to 6-Hz seizures in mice, whereas the fiber-deficient commercial infant formula KD3:1 increases susceptibility to 6-Hz seizures. Correspondingly, the protective KD4:1 and MCT2.5:1 induce several shared metagenomic alterations in the gut microbiome, which are not seen with KD3:1. In particular, KD4:1 and MCT2.5:1, but not KD3:1, reduce representation of select genes related to carbohydrate degradation, which were significantly associated with the presence of dietary fiber and similarly induced by fiber supplementation to a cultured infant gut microbial community. Adding a fiber mixture to the KD3:1 to match levels present in KD4:1 and MCT2.5:1 restores seizure protection in mice. Moreover, supplementing the fiber mixture to the already protective KD4:1 infant formula further enhances seizure resistance in mice.

Only a few small human studies have tested the effects of different medical KD regimens on seizure reduction, reporting no significant differences between the MAD, MCT, and LGIT diets relative to the classic KD in controlling seizures in children with refractory epilepsy[25,26,28,29]. However, none of these examined the role of fiber or any specific dietary constituents on patient responses to KD therapy. A cross-sectional study of 150 epileptic individuals reported insufficient intake of fiber, among several other vitamins and minerals, and that patients with low intake of vegetables exhibited greater likelihood of uncontrolled seizures[49]. When considering specific macronutrients and micronutrients that distinguish patients with controlled and uncontrolled seizures, percent intake of fiber was the closest to statistical significance (reported $p = 0.05$). In addition, a human study of KD therapy in children with refractory epilepsy reported changes in 29 metagenomic pathways, including the reduction of seven pathways involved in carbohydrate metabolism and fermentation such as fructooligosaccharides (FOS) and raffinose utilization, sucrose utilization, glycogen metabolism, lacto-N-biose I and galacto-N-biose metabolic pathway; lactate, pentose phosphate pathway; and formaldehyde assimilation: ribulose monophosphate pathway[16]. Further research is needed to explore the role of dietary fiber and specific dietary nutrients on the clinical efficacy of KD formulations for treating refractory epilepsy.

Dietary fibers are resistant to digestion by the host and specifically fermented by gut bacteria that together encode hundreds of glycoside hydrolases with varying specificity for different fiber types[50]. As such, not only does the gut microbiome degrade fiber, it also responds to and is shaped in composition and function by dietary fiber. We find that supplementing mice with the SCFAs butyrate, propionate, and acetate, as common microbial end-products of fiber fermentation, fails to phenocopy the beneficial effects of fiber supplementation on potentiating seizure protection in mice fed the KD4:1. This may align with prior human studies reporting that epilepsy is associated with deficient levels of SCFA-producing bacteria, which are further reduced by KD therapy to promote seizure control[16,22,51]. Beyond SCFAs, several other carboxylic acid metabolites, neurotransmitters, vitamins, and bile acids are also modulated by fiber fermentation[52,53]. This suggests that the fiber effects on seizure resistance may not be mediated by common SCFAs, but rather by other non-SCFA metabolites generated by fiber fermentation or indirect effects of fiber fermentation on the microbiome and host. Indeed, alterations in the gut microbiome are increasingly implicated in risk for epilepsy and seizure responsiveness to the KD across several humans studies[14–19].

Findings from animal models establish proof-of-principle that KD-induced alterations in the gut microbiome contribute to seizure resistance[12,20–23], suggesting that differential effects of dietary formulations on the gut microbiome may lead to variation in seizure protection. By screening various dietary factors that distinguish the KD infant formulas, including fat ratio, fat source, fat saturation, and carbohydrate type, on a model human infant microbial community, we find that addition of fiber to a diet-based synthetic culture media elicits substantial shifts in microbial metagenomic profiles. Many key metagenomic features seen in response to fiber supplementation in the in vitro system are consistent with those seen in the gut microbiome of seizure-protected mice fed fiber-containing KD4:1 and MCT2.5:1, suggesting direct interactions between dietary fiber and the microbiome that are effectively modeled in simplified microbial culture systems.

Different fiber types and sources can vary greatly in their chemical structure, fermentability, and effects on the gut microbiome[54]. We further expanded our in vitro screening approach to include 13 different soluble or insoluble fiber types and sources, as supplemented directly into the commercial KD4:1 infant formula (rather than a diet-based synthetic culture medium). By using key fiber-associated metagenomic features to stratify microbial responses to the 13 fiber conditions, we identified a specific subset of fibers that potentiate the seizure protective effects of the KD4:1 in mice. This subgroup, including fiber mix (inulin, FOS, gum arabic, cellulose), FOS alone, and orange fiber, is characterized by metagenomic alterations in pathways related to preQ0 biosynthesis, L-alanine biosynthesis, sucrose degradation, and partial TCA cycle. (i) PreQ$_0$ is a deazapurine nucleoside with reported antibiotic, anticancer, antineoplastic, and antiviral properties[55,56]. In mice that exhibited seizure resistance in response to transplantation of the clinical KD-induced human microbiota, microbial preQ$_0$ biosynthesis was associated with alterations in hippocampal expression of genes related to neuron generation and migration protection[20]. (ii) L-alanine is an essential amino acid that is modulated by ketosis[57] and regulates the function of glutamatergic neurons and astrocytes[58]. L-alanine levels were diminished significantly in the cerebrospinal fluid of children after four months of KD therapy[59], and genes related to L-alanine metabolism were elevated in imputed microbial metagenomic pathways from epileptic individuals relative to healthy controls[60]. Alterations in L-alanine biosynthesis could result in differential levels of 2-oxoglutarate, which is involved in the production of pyruvate and glutamate. (iii) Microbial sucrose utilization is a carbohydrate pathway reduced after KD therapy in humans[16], likely due to the low availability of carbohydrates in the diet. This specific pathway, sucrose degradation IV, is mainly encoded by *Bifidobacterium* species and shunts β-D-fructofuranose-6-phosphate to produce acetate, lactate, formate, and acetyl-coA[61], which aligns observed fiber-induced reductions in *B. breve* (Supplementary Fig. 11). (iv) In bacteria, the TCA pathway fuels aerobic respiration, wherein acetyl-CoA is converted to intermediate organic acids such as citrate, 2-oxoglutarate, and succinate. 2-oxoglutarate also known as α-ketoglutarate is an important intermediate in the TCA cycle and is known to regulate glutamate and GABA levels in the brain[62]. Cecal α-ketoglutarate levels did not differ in mice consuming KD4:1 supplemented with fiber mix, but its reduced product 2-hydroxyglutarate was significantly elevated (Supplementary Fig. 10b). How microbial levels of 2-oxoglutarate and other intermediate metabolites from the TCA cycle, may affect the brain to alter anti-seizure susceptibility is unknown. During KD therapy, patients supplemented with oral citrate as an alkalizing agent prevented metabolic acidosis without affecting the 7-month efficacy rates[63]. Similarly, KD has been shown to affect the succinate levels through succinate dehydrogenase activity in rodent models of aging[64] and through effects on mitochondrial respiration to restore ATP production[65]. (v) Peripheral glutamine is imported by the brain and contributes to central metabolism of glutamate versus GABA, which has been implicated in the anticonvulsant effects of KD[12,66]. Microbial genes related to glutamate and glutamine biosynthesis were positively associated with the presence of dietary fibers in clinical KDs and

negatively associated with fiber-deficient seizure susceptible group KD3:1 (Supplementary Fig. 2d, e). We also observed reduced cecal L-aspartate levels with fiber mix supplementation (Supplementary Fig. 10b), which could be related to reduced glutamate transamination to aspartate[66]. High levels of L-aspartate were reported in the cerebral cortex of epileptic individuals[67]. Decreased brain aspartate levels were observed in mice fed KD for 3 days[68]. Overall, these results suggest that increases in microbial biosynthetic pathways for preQ$_0$, L-alanine, and glutamate and glutamine biosynthesis, and reductions in microbial carbohydrate metabolism, may contribute to diet-induced seizure resistance. Further research is needed to determine whether there are causal links between these particular microbial functions and seizure protection.

Altogether, results from this study reveal that nuanced differences in the formulation of KDs that are used to treat refractory epilepsy can lead to major differences in treatment efficacy and in the functional potential of the gut microbiome in mice. In particular, we highlight a major role for dietary fiber in restoring and potentiating the seizure protective effects of commercial KD formulas when fed to mice. We demonstrate that dietary fiber shifts key metagenomic features in both the mouse gut microbiome and a model human infant microbial community, which can be used to identify specific fiber types that potentiate the seizure protective effects of a classical KD formula in mice. Our findings align with increasing evidence that the gut microbiome modifies the anti-seizure effects of the KD and that microbiome-targeted diets can be used to shape the structure and function of the gut microbiome. However, research is needed to test whether these findings translate to the human condition and evaluate the effects of fiber in KD therapies for pediatric epilepsy. Overall, our results from our preclinical study supports the growing notion that careful consideration of dietary effects on host-microbial interactions is needed to inform the design of more effective and personalized dietary interventions for disease.

## Methods

### Mice
All mouse experiment protocols were approved by the UCLA Institutional Animal Care and Use Committee. Juvenile (4-week old) specific pathogen free (SPF), male Swiss Webster (Taconic Farms) mice were used for all animal experiments, fed standard chow (Labdiet 5010, 28.7%: 13.1%: 58.2% protein: fat: carbohydrate by calories), and housed in sterile caging under a 12 h:12 h light:dark cycle with standard temperature and humidity control. Juvenile SPF, female Swiss Webster mice were also assessed to evaluate sex as a biological variable.

### Dietary treatment
Experimental animals were fed commercially available KD infant formulas (KetoCal, Nutricia North America, Fig. 1a, Supplementary Data 1) or a popular commercially available, standard infant formula as control diet (Abbott Nutrition, Fig. 1a, Supplementary Data 1) for 7 days. For liquid diet paradigm, 90 g of powder formula or 90 mL of liquid formula was mixed with 600 mL water at 60 °C. Before adding to cages, the diet solution was brought to 1 L and each cage containing 3–4 mice was supplemented with liquid diets in water bottles. The water bottles were filled with liquid diets and the cages were changed every 1–2 days. For the solid diet paradigm, 90 g of powder formula was mixed with 600 mL water and dehydrated using a food dehydrator (CASORI). The diets were administered in sterile petri dishes ad libitum and cages were provided with standard sterile water. For the pasted diet paradigm, 90 g of powder was mixed with water and administered as a paste in sterile petri dishes.

For fiber supplementation experiments, 5 g of individual fiber or fiber mixture (fructooligosaccharides (FOS, >90%, quality level (ql) = 200), inulin from chicory (Sigma-Aldrich or MPBiomedicals >90%, ql = 200), microcrystalline cellulose (ql = 100), and gum arabic (from

acacia tree, ql = 200) from Sigma-Aldrich, mixed at 1:1 (w/w)) was added to 90 g of KD formula prior to administering as a paste as described above. For SCFA supplementation, we considered the following concentrations as reference values for SCFAs reported in SPF mice fed standard chow containing 15% fiber: acetate (67.5 mM), propionate (25 mM) and butyrate (40 mM)[69]. To model 5% fiber content present in KD4:1, we therefore administered sodium acetate (22.55 mM), sodium propionate (8.33 mM) and sodium butyrate (13.35 mM) in sterile drinking water. For paste diets, 1:10 amount of the SCFA mixture were mixed with water and added to the powder diets (90 g) to keep the same concentrations above. As a negative control, sodium chloride (NaCl) was supplemented to match amounts in SCFA salts in water and in diet (44 mM).

### 6-Hz psychomotor seizure assay
The 6-Hz psychomotor seizure test was conducted as previously described with modification[12]. One drop (∼50 ul) of 0.5% tetracaine hydrochloride ophthalmic solution was applied to the corneas of each mouse 10–15 min before stimulation. Corneal electrodes were coated with a thin layer of electrode gel (Parker Signagel). A current device (ECT Unit 57800, Ugo Basile) was used to deliver current at 0.3 s duration, 0.2 ms pulse-width and 6 pulses/s frequency. CC50 (the intensity of current required to elicit seizures in 50% of the experimental group) was measured as a metric for seizure susceptibility. Pilot experiments were conducted to identify 28 mA as the CC50 for SPF wild-type Swiss Webster mice when they are on liquid and solid diet and 44 mA when they are on paste diet. Each mouse was seizure-tested only once, and thus $n$ = 14–16 mice were used to adequately power each experimental group. Power calculation was performed using R package pwr v1.3 using seizure thresholds for mice fed KD vs CD reported in previous studies[12,20]. The results indicated a sample size of 3–11 to meet 0.8 power and 0.001 significance level. 28 or 44 mA currents were administered to the first mouse per cohort, followed by fixed increases or decreases by 2 mA intervals. Mice were restrained manually during stimulation and then released into a new cage for behavioral observation. Locomotor behavior was recorded using a camera and quantitative measures for stunned fixture, falling, tail dorsiflexion (Straub tail), forelimb clonus, eye/vibrissae twitching, and behavioral remission were scored manually. Latency to exploration (time elapsed from when an experimental mouse is released into the observation cage (after corneal stimulation) to its normal exploratory behavior) was scored manually with an electronic timer. Mice were blindly scored as protected from seizures if they did not show seizure behavior and resumed normal exploratory behavior within 10 s. Seizure threshold (CC50) was determined as previously described[70], using the average log interval of current steps per experimental group, where sample n is defined as the subset of animals displaying the less frequent seizure behavior. Data used to calculate CC50 are also displayed as latency to explore for each current intensity, where n represents the total number of biological replicates per group regardless of seizure outcome.

### Fecal shotgun metagenomics
Frozen stool samples collected from one mice per cage post-dietary treatment were subjected to DNA extraction using the ZymoBIOMICS DNA Miniprep kit (Zymo), with bead beating used to lyse cells. Briefly, the samples were transferred into PowerBead tubes containing lysis solution and bead beaded at maximum speed for 1 min five times with 1 min of ice incubation in between cycles. The rest of the protocol followed the manufacturer's instructions. The DNA was eluted in 60 μL elution buffer provided by the kit. Purified DNAs were sent to Novogene Corporation Inc for paired end (PE) metagenomic sequencing. Sequencing was performed on the Illumina NovaSeq platform with PE reads of 150 bp for each sample averaging around 3GB data. Raw reads were subjected to kneaddata to remove contaminants, and trimmed forward reads were used for functional and taxonomic analysis.

Metagenomic data was analyzed using HUMAnN3[71] (v3.8) and MetaCyc database (uniref50_201901b) using diamond v2.1.8 to profile gene families and pathway abundance. MetaPhlAn4 (v4.0) was used for metagenomic taxonomic profiling[72]. α-diversity indexes for taxonomic profiling were determined by Shannon's index, richness, and Pielou's Evenness using vegan v2.6-4 in R. For β-diversities, calculate_diversity.R script were run within the MetaPhlAn4 based on the relative abundances. For Unifrac distances, mpa_vOct22_CHOCOPhlAnSGB_202212.nwk was used for SGB-level phylogenetic tree as reference. R packages tidyverse v2.0.0, vegan v2.6-4, and phyloseq v1.38.0 was used for principal coordinate analysis (PCoA) of taxanomic distribution. Alterations in microbial diversity were assessed using PERMANOVA with adonis2 with 999 permutations from the vegan package in R. File2meco R package was used for MetaCyc pathway hierarchical classification[73]. Pathway abundances were normalized within the HUMAnN3 package to obtain counts per million (CPM) abundances (Supplementary Data 8) and MaAsLin 2.0[74] with log transformation was used to assess significant pathway associations between dietary treatments with an adjusted $p$ value ($q$ value) cutoff of 0.05, where indicated in the figure by asterisk.

## Beta-hydroxybutyrate (BHB) measurements

Blood was collected via a capillary tube from the medial canthus of the eye, allowed to clot 30 min at room temperature, and spun through SST vacutainers (Becton Dickinson) at $15,000 \times g$ for 90 s for serum separation. Samples were immediately snap frozen in liquid nitrogen and stored at −80 °C until further processing. BHB levels were quantified by colorimetric assay according to the manufacturer's instructions (Cayman Chemical).

## Bacterial strains and culturing

The following strains were selected to represent the human infant gut microbiome, based on their high relative abundance in their respective phyla and prevalence at >1% relative abundance across the study population[40,41] (Supplementary Fig. 3a). Type strains were obtained either from ATCC or DSMZ collection and propagated as instructed: *Bifidobacterium longum* subsp. *infantis* DSM 20088, *Bifidobacterium longum* subsp. *longum* ATCC BAA-999, *Bifidobacterium breve* DSM 20213, *Bacteroides fragilis* ATCC 25285, *Bacteroides vulgatus* ATCC 8482, *Enterococcus faecalis* ATCC 19433, *Clostridium perfringens* ATCC 13124, *Escherichia coli* K-12 ATCC 10798, *Klebsiella pneumoniae* subsp. *pneumoniae* ATCC 13883. The cultures were routinely grown anaerobically in their respective media and temperature (Supplementary Data 3). The growth of species were tested on a rich complex medium[42] for 24 h to confirm stable relative abundances over the duration of anaerobic culture, as confirmed by cfu plating and qPCR (Supplementary Data 2, Supplementary Fig. 3b).

## In vitro batch culture fermentations

Synthetic KDs with different ratios, fat and carbohydrate source were prepared using sunflower oil (Baja Precious, 93%), vegetable shortening (Crisco, 100%), palm oil (Ökonatur, 100%), soy lecithin (Modernist Pantry, 82%), linoleic acid (Sigma-Aldrich, 99%), and Medium Chain Triglycerides (MCT, Nutriticia, 50%) as fat sources, whey protein isolate (Bulk Supplements, 90%) as protein source, and lactose (modernist pantry, 99%) and dietary fiber mixture of fructooligosaccharides (Sigma-Aldrich, >90%), inulin from chicory (Sigma-Aldrich, >99%), crystalline cellulose (Sigma-Aldrich, >99%), and gum arabic from Acacia Tree (Sigma-Aldrich, >99%) as carbohydrate sources. Additionally, for fiber supplementation fermentation experiments, wheat (Vitacel, WF600), pea (Vitacel, EF100), potato (Vitacel KF150 PLUS), and apple (Vitacel, AF 401) fiber from J. Rettenmaier USA LP, orange (citrus) fiber from Citri-Fi Naturals (100%), oat (NuNaturals, 100%), acacia (Nutricost organic, 100%), and psyllium husk (It's just, 100%) were used. The powders were ultraviolet (UV)-sterilized and

confirmed to be sterile by aerobic and anaerobic culture. They were then mixed with simulated saliva solution, gastric solution, and intestinal fluid as described in INFOGEST model[75] without enzymatic solution (Supplemental Data 4) to simulate the gastric and intestinal bolus entering to the colon and was subjected to an in vitro batch culture fermentation. The representative bacterial strains were mixed in a minimal media at the dilution factor (1:100) needed to achieve a ratio of 21% Actinobacteria, 14% of Bacteroidetes, 28% of Firmicutes, and 37% of Proteobacteria, reflective of relative abundances seen in a typical infant gut[40,41] (Supplemental Data 3). Species that comprise Actinobacteria, Bacteroidetes and Firmicutes were mixed at 1:1 ratio, whereas Proteobacteria consists of 57 % of *Escherichia coli* and 42% of *Klebsiella pneumoniae*. The bacterial mixture was then mixed with each diet bolus (1:1 v/v) and subjected to 24-h anaerobic fermentation at 37 °C. After 24 h, the bacterial pellets were separated from the media by centrifugation at $16,000 \times g$ for 5 min and stored at −80 °C until further analysis. The pellets from pre-fermentation were also collected as a control.

## Bacterial quantification via qRT-PCR

Total DNA was extracted from the pellets collected after fermentation, following standard procedures for the ZymoBIOMICS DNA Miniprep kit. The microbial composition was determined using quantitative RT-PCR with species-specific primers[76–82] and respective qPCR conditions (Supplementary Data 5). DNA extracted from individual overnight cultures was used to generate a standard curve. The copy numbers for each sample were calculated based on the standard curve and normalized to DNA concentration of the original sample. Absolute quantification of growth after anaerobic culture of each sample was determined by subtracting the pre-fermentation quantities and presented as log values. Any species that exhibited negative values after subtraction were regarded as zero or no growth. Data are presented in bar plots as a mean of each bacteria. PCoA plots were created using cmdscale from the distance matrix created using Euclidean distances in vegan package in R. For conditions KD4:1 supplemented with fiber mix or the individual fibers, an outlier analysis was performed by outliers v0.15 package in R and qPCR values were removed from further analyses.

## Production of synthetic metagenome reads and synthetic metagenome analysis

The genome fastq files for each species were obtained from ATCC.org. Bowtie2-build v2.3 was used to build indexes for reference genomes. Open source BBMap v38.94 randomreads.sh plugin was used to randomly produce paired reads at 150 bp length from each genome based on the qPCR absolute quantification multiplied by a million. For each sample, the reads that were produced from each genome were concatenated and between 50–80 million metagenomic reads were created. Metagenomes were analyzed using HUMAnN3[71] (v3.9) and MetaCyc database (uniref50_201901b) using diamond v2.1.8 and significant pathway associations were determined with MaAsLin2 package in R as described above.

## Short chain fatty acid (SCFA) measurements

The LC-MS analysis of SCFA in cecum and serum was performed by the Analytical Phytochemical Core at UCLA with modifications based on previously published protocols[83,84]. Cecal contents were collected post-behavioral testing from terminal mouse dissections, immediately snap frozen in liquid nitrogen and stored at −80 °C until analysis. Blood was collected via retro-orbital sampling as described above. 50 mg of cecal contents were homogenized in 200 µL PBS (Bead Mill, Fisher Scientific), vortexed and sonicated in ice water for 10 min. The lysate was then centrifuged at $14,000 \times g$ for 10 min at 4 °C and the supernatant was collected. 30 µl serum or cecal lysate supernatant were mixed well with 30 µl isopropyl alcohol with isotope labelled internal

standards (Acetic-d3, Propionic-d5 and Butyric-d7 5 μg/mL each), pre-cipitated at −20 °C overnight, and centrifuged at 14,000 × g for 10 min at 4 °C. The supernatant was derivatized prior to LC-MS following previously published instructions[83,84]. The LC-MS analysis was per-formed on an Agilent Zorbax SB-C18 2.1 × 150 mm column using the TSQ Quantum (Thermo-Finnigan) LC–ESI-MS/MS system at negative mode. Quantification was achieved by using Xcalibur data system. Calibration curves were built by fitting the analyte concentrations versus the peak area ratios of the analyte to isotope labeled internal standards. The peak area ratios target analyte to isotope-labeled internal standards in the samples were used to calculate the concentrations.

### Targeted polar metabolite analysis

Polar metabolites were extracted based on previous publications with modifications[85–87]. Briefly, 50 mg of frozen cecal content was added to a pre-cooled bead beading tube containing 100 mg of 0.1 mm glass beads (Qiagen), 100 mg of 212–300 μm acid washed glass beads (Sigma-Aldrich), and 2 of 4 mm acid washed silica beads (OPS Diag-nostics). 1000 μL of the extract solution (acetonitrile (Thermo-Scientific Chemicals, HPLC Grade)–methanol (Sigma-Aldrich, HPLC grade)–water (Fisher Chemical, HPLC Grade) = 2:2:1) was added into the tubes and homogenized in by bead beating (BioSpec Products) for 6–10 rounds of 30 s at max speed with 5 min incubations on −20 °C in between rounds until no particles were observed. Homogenate was incubated at −20 °C overnight, then centrifuged at 16,000 × g for 15 min at 4 °C and stored at −20 °C until analysis. The polar metabolite separation was done on HILIC column using a UPLC system equipped with mass spectrometer (Q-Exactive, Thermo Scientific) at the UCLA Metabolomics Center as previously described[88,89]. Peaks were aligned among all samples and assigned identities using exact mass and retention time based on core's in-house database. Peaks were quanti-fied by area under the curve integration and metabolites were filtered if less than 50% of the replicates had measurements.

### Statistical analysis and reproducibility

All statistical analyses were conducted using R version 4.1.2. Power calculation was performed using R package pwr v1.3 to determine the sample sizes. Data for box-and-whisker plots were plotted as median with first and third quartiles. Data for parametric data sets was ana-lyzed using one-way ANOVA with Bonferroni adjustment between multiple groups. For differences between two sample conditions, parametric datasets were analyzed using a two-sided Welch's t-test and non-parametric data sets using Wilcoxon signed rank test. For non-parametric distributions with more than two groups, data was ana-lyzed by Kruskal-Wallis with Dunn's test. For PCoA plots, the distance matrix created within vegan package was initially subjected to beta-disper and permutest for multivariate homogeneity of groups dis-persions (variances), then PERMANOVA with adonis2 (one-sided by default) with 999 permutations was used to determine statistical dif-ferences between groups. Significant differences from the tests were denoted as follows: *$p < 0.05$, **$p < 0.01$, ***$p < 0.001$. Notable non-significant differences were denoted as n.s.

All data are representative of and/or include biological replicates. Mice were randomly assigned into different dietary conditions. A blin-ded researcher was assigned to re-review the seizure behavior. All metagenomics and metabolite analysis were performed on repre-sentative samples randomly collected from each cage for each experi-mental condition. No data were excluded from the animal experiments. In vitro bacterial fermentation experiments, bacterial colonies were randomly selected as biological replicates and performed at least two separate experiments performed on different days. An outlier analysis was performed for fiber supplemented KD4:1 screening on model infant microbial community as described above.

### Reporting summary

Further information on research design is available in the Nature Portfolio Reporting Summary linked to this article.

## Data availability

Raw and processed data from taxonomic and metagenomic profiling from mice, and associated metadata are presented in Supplementary Data Tables S7 and S8 are available online through the NCBI Sequence Read Archive (SRA) repository at SRA: PRJNA1184197. Raw an pro-cessed metagenomic data created synthetically and associated meta-data are presented in Supplementary Data Tables S9, and S10 and available online through QIITA[90]: 15789. Raw data from polar meta-bolite analysis is presented in Supplementary Data Table S11. The data used to generate the figures are included in source data files or sup-plementary data tables. Source data are provided with this paper.

## Code availability

No original code or software was produced for this paper. The codes and software used for this paper are listed in the "material and meth-ods" section.

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

## Acknowledgements

We thank Joyce H. Matsumoto and Beck Reyes at the UCLA's Mattel Children's Hospital for valuable discussion on clinical KD treatments, Jieping Yang and Ru-Po Lee at the UCLA Human Nutrition Analytical Phytochemical Core Facility for technical support with LC-MS for SCFA measurements, and Johanna ten Hoeve-Scott at the UCLA Metabolomics Center for polar metabolite analysis. This work was supported by NINDS grant #R01NS115537 (E.Y.H.).

## Author contributions

E.Ö., K.B.Y., G.R.L., J.P., A.L.R., and E.Y.H. conceptualized and planned bacterial and mouse experiments. E.Ö., K.B.Y., L.D., K.L., J.H., and M.A. performed bacterial experiments. E.Ö., K.B.Y., and G.R.L. performed mouse experiments. E.Ö. performed metagenomics, metabolite extractions, and analyzed data. E.Ö. and E.Y.H. wrote the manuscript. All authors contributed to final manuscript.

## Competing interests

The findings are the subject of UCLA provisional patent application US 63/677,089. The authors declare no other competing interests.
