## [Transparent Peer Review file · Nature Communications]

Dietary fiber content in clinical ketogenic diets modifies the gut microbiome and seizure resistance in mice

Corresponding Author: Dr Elaine Hsiao

Version 0:

Reviewer comments:

Reviewer #1

(Remarks to the Author)

The manuscript by Özcan et al. investigates in detail the components of commercially available infant KD formulations for their seizure protective effects in an established epilepsy mouse model. They find that rather the dietary fiber source than the fat content/source modulate seizure thresholds and that this effect is likely mediated through the functions of the gut microbiota feeding on these fibers. It is a significant and highly novel finding with clinical implications. The conclusions are based on sound experimental design and results.

However, I have a few major concerns. Please, see below:

- 1) Microbiome data analysis: I cannot find any mentioning of any log-ratio transformation of the dataset to deal with the compositional nature of the dataset. This is very important to do prior to statistical analysis to avoid many false positives. Especially in experiments drastically changing the energy sources of the gut microbes. E.g. in a fiber-deprived KD the total bacterial load in the gut may be lower than in the mice receiving fibers in their diet and different bacterial loads bias the relative abundances heavily. More information can be found here: <https://www.frontiersin.org/journals/microbiology/articles/10.3389/fmicb.2017.02224/full> especially Fig. 1b. Therefore, the statistical analyses need to be repeated after appropriate transformation. I suggest CLR. Using clr-transformation and Euclidean distances for beta diversity results in Aitchison distances which are superior to Bray-Curtis dissimilarity metrics used in the manuscript (<https://link.springer.com/article/10.1023/A:1007529726302>). Therefore, they should be replaced.
- 2) I was surprised to see any KD decreasing seizure thresholds. While this is in line with the findings in the manuscript using a mouse model, I am confused, as KD3:1 is used as treatment in patients. What is the data for this formulation in patients? How effective is it? I cannot imagine it increases seizures. In that case it should have been discontinued. Therefore, I am hesitant about the translatability of these findings from mice to patients. Please, discuss the clinical effect of KD3:1. I also believe (but I admit, I haven't seen any data on this) that patients in general decrease their fiber consumption when starting KD as there is not much room for carbohydrates that usually come with the fibers in whole foods. How come, KD is so effective in patients still? Therefore, there should also be more emphasis on the fact that these findings are relevant in mice but potentially not in patients throughout the manuscript.
- 3) Based on the results presented and conclusions drawn, it seems to come down to a protective fiber mix modulating seizures. Therefore, I wonder. Do we even need a KD? An experiment measuring seizure thresholds in mice on CD vs CD+fiber mix should be performed!
- 4) How are these findings related to your previous findings, e.g. the gamma-glutamylated amino acids?

Minor points:

Please, do not use the term "biomarker". It was not thoroughly enough investigated whether the metagenomic signatures can serve as biomarkers.

Line 624: "donated" should be "denoted".

Line 867: what is the color scale representing?

In summary, I think the study is interesting and important! If my concerns can be addressed appropriately, I can recommend it for publication.

Reviewer #2

(Remarks to the Author)

The noteworthy results are that different formulations of the clinical ketogenic diets, and dietary fiber content in particular, differentially impact seizure outcome in mice, likely through modification of the gut microbiome.

The paper is well written and the results are important and novel.

Methodology is appropriate. However, more details on the animal experiments should be provided, in the methods for the work to be reproduced. In terms of how long the dietary treatments were administered, it is stated that 1 week of feeding was selected—was this duration adequate to modulate the gut microbiome? Clarify how many samples were taken during the intervention period, and analysed at each time point. Details of the power calculation to show that $n=14-16$ mice were used to adequately power each experimental group should be given.

They found that KD3:1 infant formula yielded decreased seizure thresholds compared to all other groups, including CD-fed controls, but this was alleviated by adding the fiber mix.

The authors directly added SCFA to further ask whether fiber supplementation promotes seizure resistance via SCFAs. Additionally, direct measurement of SCFA in colon/cecal contents and in feces following fiber supplementation could yield data to answer this question. Additional metabolomics analysis to compare the metabolomes from gut samples from animals on the different diets would shed further light on which metabolites were enhanced or reduced and would add value to this study.

The addition of SCFAs directly to the KD4:1 infant formula did not yield the same results as feeding the dietary fiber. What levels of SCFA were reached in the colon of fiber fed and SCFA fed mice? I did not see these data reported in the manuscript.

Was it confirmed that the directly fed SCFA reached similar levels in the colon as were achieved by feeding dietary fiber? If not, the conclusion that the anti-seizure effects of the fiber containing KD4:1 were not mediated through SCFA was not fully demonstrated.

The model human infant microbial community included the following strains, based on their high relative abundance in their respective phyla and prevalence at $>1\%$ relative abundance across the study population.

Type strains from either from ATCC or DSMZ collection
Bifidobacterium longum subsp. infantis DSM 20088,
Bifidobacterium longum subsp. longum ATCC BAA-999,
Bifidobacterium breve DSM 20213,
Bacteroides fragilis ATCC 25285,
Bacteroides vulgatus ATCC 8482,
Enterococcus faecalis ATCC 19433,
Clostridium perfringens ATCC 13124,
Escherichia coli K-12 ATCC 10798,
Klebsiella pneumoniae subsp. pneumoniae ATCC 13883.

Were they each tested against the other, to ensure that no strain produced antimicrobial activity against others in the community?

Version 1:

Reviewer comments:

Reviewer #1

(Remarks to the Author)

I want to thank the authors for thoroughly addressing my comments. I have nothing more to add.

Reviewer #2

(Remarks to the Author)

I am satisfied that all my comments have been addressed in the revision of this manuscript.

We appreciate the valuable comments and suggestions provided by the Editorial team and Reviewers. We are confident that we have thoroughly addressed all the points raised through additional experiments, analyses, and textual revisions to enhance the robustness of our study.

Reviewer #1 (Remarks to the Author):

The manuscript by Özcan et al. investigates in detail the components of commercially available infant KD formulations for their seizure protective effects in an established epilepsy mouse model. They find that rather the dietary fiber source than the fat content/source modulate seizure thresholds and that this effect is likely mediated through the functions of the gut microbiota feeding on these fibers. It is a significant and highly novel finding with clinical implications. The conclusions are based on sound experimental design and results.

We thank the Reviewer for acknowledging the novelty and rigor of our study.

However, I have a few major concerns. Please, see below:

1) Microbiome data analysis: I cannot find any mentioning of any log-ratio transformation of the dataset to deal with the compositional nature of the dataset. This is very important to do prior to statistical analysis to avoid many false positives. Especially in experiments drastically changing the energy sources of the gut microbes. E.g. in a fiber-deprived KD the total bacterial load in the gut may be lower than in the mice receiving fibers in their diet and different bacterial loads bias the relative abundances heavily. More information can be found here: <https://www.frontiersin.org/journals/microbiology/articles/10.3389/fmicb.2017.02224/full> especially Fig. 1b. Therefore, the statistical analyses need to be repeated after appropriate transformation. I suggest CLR. Using clr-transformation and Euclidean distances for beta diversity results in Aitchison distances which are superior to Bray-Curtis dissimilarity metrics used in the manuscript (<https://link.springer.com/article/10.1023/A:1007529726302>). Therefore, they should be replaced.

We thank the Reviewer for their insightful suggestion. Our original taxonomic profiling used outputs from MetaPhlan that are based on relative abundances with no log transformation. We have now followed the Reviewer's suggestion to calculate Aitchison distances. To do so, we used the MetaPhlan package utilities and calculate_diversity.R script, which was also used to for other beta-diversity calculations. We performed statistical analysis using PERMANOVA, and added a PCoA plot showing Aitchison distances in **Figure 2b**. The results similarly show that mice fed different clinical KD formulas exhibit altered composition of the fecal microbiota.

Fig. 2b. Principal coordinates analysis (PCoA) of Aitchison distance (left) based on fecal metagenomic sequencing data after dietary treatment. (PERMANOVA, n = 4 cages/group).

We added the following text for clarification in the Methods section line 575-576:

“For beta-diversity, the calculate_diversity.R script was run within MetaPhlan4 based on microbial relative abundances.”

We also added the text below in the Results section lines 150-152.

“Beta-diversity analysis of the gut microbiota based on Aitchison distance showed that KD3:1 clustered distinctly from CD controls and KD4:1 along PCoA2 and from MCT2.5:1 along PCoA1 (PERMANOVA, $p=0.001$ $R^2=0.46$, Fig. 2b).”

For metagenomic profiling, pathway abundances were already normalized to cpm abundances and for statistical analysis in MaAsLin2, they were subjected to log transformation. We added the following details in the Methods section in lines 581-585 for clarification.

“Pathway abundances were normalized within the HUMAnN3 package to obtain counts per million (CPM) abundances. MaAsLin 2.0⁷⁴ with log transformation was used to assess significant pathway associations between dietary treatments with an adjusted p value (q value) cutoff of 0.05, where indicated in the figure by asterisk.”

2) I was surprised to see any KD decreasing seizure thresholds. While this is in line with the findings in the manuscript using a mouse model, I am confused, as KD3:1 is used as treatment in patients. What is the data for this formulation in patients? How effective is it? I cannot imagine it increases seizures. In that case it should have been discontinued. Therefore, I am hesitant about the translatability of these findings from mice to patients. Please, discuss the clinical

effect of KD3:1. I also believe (but I admit, I haven't seen any data on this) that patients in general decrease their fiber consumption when starting KD as there is not much room for carbohydrates that usually come with the fibers in whole foods. How come, KD is so effective in patients still? Therefore, there should also be more emphasis on the fact that these findings are relevant in mice but potentially not in patients throughout the manuscript.

We agree with the Reviewer that this was an unexpected result that is worth further discussion in the manuscript. We unfortunately do not know the formula brand's rationale to include fiber in the KD4:1 and MCT2.5:1, but not KD3:1. We have contacted them for more information, but have not received a response. In the literature, there are no existing studies that have methodically tested fiber for its causal effects on KD-induced seizure resistance. There are also no published studies that have specifically tested for differential effects of these particular infant formulas on epilepsy. We believe that our study will be a valued addition in this regard.

We have contacted our clinical collaborators at the UCLA Mattel Children's hospital to learn more about their experiences with fiber supplementation in epileptic children on KD therapy. They routinely use a single formulation of KD infant formula (KD4:1) as a base, and when necessary, they add standard formula to achieve lower fat ratios. This avoids switching to different manufacturer formulations, and is more easily tolerated and maintained by patients. Fiber supplements are only prescribed in the event that gastrointestinal issues like constipation or loose stool are reported. Anecdotally, they do not observe striking differences in symptoms with fiber supplementation. However, improvements in seizure outcome may be expected with alleviation of gastrointestinal issues. An additional explanation could be that fiber supplementation increases digestion times and attenuates glucose responses, mimicking the low glycemic index treatments (LGIT) for epilepsy. They note that in the pediatric population, initiating KD therapy does not necessarily result in a decrease in fiber consumption relative to their typical diet, especially for children with restricted food interests or fed infant formulas that do not contain fiber or human milk oligosaccharides. Overall, we believe that the sparsity of information on these questions signals a need for continued research that evaluates roles for specific dietary formulations on the efficacy of KD therapies. We agree with the Reviewer that with animal studies, translatability to humans is always an uncertainty.

In lines 402-403 of the discussion section, we acknowledge the lack of existing studies that have tested the effects of fiber.

In lines 482-496 of the discussion section, we emphasize at multiple points that our findings are specifically from mice. We also now explicitly state that work is needed to assess whether they are translatable to humans:

“ Altogether, results from this study reveal that nuanced differences in the formulation of KDs that are used to treat refractory epilepsy can lead to major differences in treatment efficacy and in the functional potential of the gut microbiome **in mice**. In particular, we highlight a major role for dietary fiber in restoring and potentiating the seizure protective effects of commercial KD formulas when fed to mice. We demonstrate that dietary fiber shifts key metagenomic features in both the mouse gut microbiome and a model human infant microbial community, which can

be used to identify specific fiber types that potentiate the seizure protective effects of a classical KD formula in mice. Our findings align with increasing evidence that the gut microbiome modifies the anti-seizure effects of the KD and that microbiome-targeted diets can be used to shape the structure and function of the gut microbiome. However, research is needed to test whether these findings translate to the human condition and evaluate the effects of fiber in KD therapies for pediatric epilepsy. Overall, the results from this preclinical study support the growing notion that careful consideration of dietary effects on host-microbial interactions is needed to inform the design of more effective and personalized dietary interventions for disease.

3) Based on the results presented and conclusions drawn, it seems to come down to a protective fiber mix modulating seizures. Therefore, I wonder. Do we even need a KD? An experiment measuring seizure thresholds in mice on CD vs CD+fiber mix should be performed!

We thank the Reviewer for raising this important question. As suggested, we have now assessed 6-Hz seizure threshold in mice fed CD+fiber mix vs. CD as a paste diet. Mice fed CD+fiber mix exhibited a modest increase in seizure threshold that was statistically significant but unlikely to be biologically significant due to the very small effect size relative to those observed with KD+fiber mix and those typically seen with KD consumption and anti-seizure medications. As such, we conclude that the fiber mix promotes resistance to 6-Hz seizures particularly in the context of KD consumption. We added the following text in the Results section lines 279-283 and the following figure in Supplementary Figure 6:

“We further investigated whether adding excess fiber to CD formula could enhance seizure protection. Fiber supplementation to CD-fed mice resulted in only a very minimal increase (CC50_CD = 48.19, CC50_CD+Fiber = 48.67, $p < 0.01$) (Supplementary Fig. 6), suggesting that fiber supplementation is more effective in the context of KD consumption.”

Supplementary Figure 6. Addition of fiber to CD does not substantially affect seizure resistance.

a. 6-Hz seizure threshold (left) and latency to exploration (right) for mice fed CD and CD+fiber mix as paste diets (left, Welch t-test, n=15-16 mice/condition, **p<0.01). Yellow line at y = 10 s represents threshold for scoring seizures.

b. Average consumption of paste diets (n=4 cages/condition; Kruskal-Wallis with Dunn's test. Data are presented as box-and-whisker plots with median and first and third quartiles).

4) How are these findings related to your previous findings, e.g. the gamma-glutamylated amino acids?

We thank the Reviewer for raising this point. In our study of the clinical KD-induced human gut microbiota (Lum et al, 2023), microbial preQ₀ biosynthesis was associated with alterations in hippocampal expression of genes related to neuron generation and migration protection. This aligns with our current finding that genes related to microbial preQ₀ biosynthesis are enriched in mice that exhibit fiber-induced seizure resistance. In our study of the KD chow-induced mouse gut microbiota (Olson et al, 2015), reductions in gamma-glutamylated amino acids were associated with increases in colonic glutamine levels and central GABA relative to glutamate levels. Although our current study did not highlight microbial pathways related to the production of the gamma-glutamylated amino acids, we observed that enrichment of genes related to glutamate and glutamine biosynthesis were positively associated with seizure protection. We have discussed these findings in the following lines in the Discussion Section

Lines 443-446:

“In mice that exhibited seizure resistance in response to transplantation of the clinical KD-induced human microbiota, microbial preQ₀ biosynthesis was associated with alterations in hippocampal expression of genes related to neuron generation and migration protection²⁰.”

Lines 467-472:

“Peripheral glutamine is imported by the brain and contributes to central metabolism of glutamate versus GABA, which has been implicated in the anticonvulsant effects of KD^{12,64}. Microbial genes related to glutamate and glutamine biosynthesis were positively associated with the presence of dietary fibers in clinical KDs and negatively associated with fiber-deficient seizure susceptible group KD3:1 (**Supplementary Fig. 2d,e**).”

Minor points:

Please, do not use the term “biomarker”. It was not thoroughly enough investigated whether the metagenomic signatures can serve as biomarkers.

Line 624: “donated” should be “denoted”.

Line 867: what is the color scale representing?

We thank the Reviewer for identifying these needs for correction. We deleted the term “biomarker”.

In line 705, we corrected “donated” as “denoted”.

We clarified the color legends in the figure legends:

“For Fig. 2e, Supplementary Fig. 2d,e, Figure 3c, Supplementary Fig. 4 and Fig 4d color scheme represents the coefficient values from MaAsLin2 within the feature table.”

Reviewer #2 (Remarks to the Author):

The noteworthy results are that different formulations of the clinical ketogenic diets, and dietary fiber content in particular, differentially impact seizure outcome in mice, likely through modification of the gut microbiome.

The paper is well written and the results are important and novel.

We thank the Reviewer for their assessment and acknowledging the novelty of our study.

1) Methodology is appropriate. However, more details on the animal experiments should be provided, in the methods for the work to be reproduced. In terms of how long the dietary treatments were administered, it is stated that 1 week of feeding was selected—was this duration adequate to modulate the gut microbiome? Clarify how many samples were taken during the intervention period, and analysed at each time point.

We thank the Reviewer for their suggestion to clarify these methodological details.

1 week for dietary treatments was adequate to modulate the gut microbiome based on the following points:

1. We have provided mouse metagenomic data in our study in Figure 2, demonstrating significant changes in the gut microbiome following 1 week of dietary treatment.
2. The 1-week duration was chosen based on findings from a previous longitudinal study, which showed that KD chow shifted the gut microbiome and conferred seizure protection within 4 days (and through 28 days). This justification is provided in our manuscript on lines 111–113:

“1 week of feeding was selected based on our prior longitudinal characterization, which indicated that KD chow shifts the gut microbiome and confers seizure protection by day 4 of treatment in mice¹²”

The fecal samples were collected after 1 week of dietary treatment as illustrated in the diagram in Figure 1. The number of samples used for each analysis is stated in each figure legend.

We have edited the Methods section of the manuscript to provide additional details:

“Fecal shotgun metagenomics

Frozen stool samples, collected from one mouse per cage post-dietary treatment, were subjected to DNA extraction using the ZymoBIOMICS DNA Miniprep kit (Zymo), with bead beating used to lyse cells.”

2) Details of the power calculation to show that n=14-16 mice were used to adequately power each experimental group should be given.

We thank the Reviewer for this suggestion. We performed power analysis based on seizure thresholds that were provided in two previous studies that used 6 Hz testing with KD and CD. Based on Lum et al, 2023, comparing conventionalized (GF-CONV) mice fed KD vs CD for 4 days (CD log dose=1.53 and KD log dose=1.64 pooled SD=0.065 to reach power=0.8 on a two-sided t-test and significant levels 0.001), we calculated that 11 mice were needed per group. Based on Olson et al 2018, comparing conventional (SPF) mice fed KD vs CD for 8 days (CD cc50 log dose=1.26, KD cc50 log dose=1.48, SD~0.01 to reach power=0.8 on a two-sided t-test and significant levels 0.001), we calculated that 3 mice were needed per group. Combining both studies, we added the following text in the Methods section on lines between 543-545:

“Power calculation was performed using R package pwr v1.3 using seizure thresholds for mice fed KD vs CD reported in previous studies^{12,20}. The results indicated a sample size of 3-11 to meet 0.8 power and 0.001 significance level.”

3) They found that KD3:1 infant formula yielded decreased seizure thresholds compared to all other groups, including CD-fed controls, but this was alleviated by adding the fiber mix.

The authors directly added SCFA to further ask whether fiber supplementation promotes seizure resistance via SCFAs. Additionally, direct measurement of SCFA in colon/cecal contents and in feces following fiber supplementation could yield data to answer this question.

This point is well taken. We performed SCFA measurements in serum and cecum samples collected from mice fed KD4 or KD4+fiber. We observed no statistically significant increases in serum and cecal levels of acetate, butyrate, and propionate following fiber fermentation. This was unsurprising to us given our prior experiences with SCFA supplementation (Coley-O'Rourke et al., 2024 and Pronovost et al., Sci. Adv, 2023), which suggested rapid absorption and metabolism of exogenously administered SCFAs. The inability to detect significant elevations in SCFA following fiber supplementation may also be due to the timing of sample collection. In this case, samples were harvested in the afternoon for mice housed on standard light cycle. This would represent 6+ hours after exposure to fiber, as mice are expected to primarily eat during the active cycle at night.

Supplementary Figure 9. There were no observable increases in serum or cecal short chain fatty acid levels in mice fed KD4:1 supplemented with fiber mix

a. Acetate, butyrate and propionate concentration in serum (left) and cecum (right) after 7 days of consumption of KD4:1 and KD4:1+fiber mix as a paste diet (n=4 cages/group, Wilcoxon signed-rank test. Data are presented as box-and-whisker plots with median and first and third quartiles.)

b. Average consumption of solid diets (left) and total SCFA concentration in serum normalized to diet consumption (n=4 cages; Wilcoxon signed-rank test. Data are presented as box-and-whisker plots with median and first and third quartiles).

We therefore added the following text in the Results section lines 301-310:

“In both liquid and solid form, SCFA supplementation failed to phenocopy effects of dietary fiber supplementation and instead yielded mice with modest reductions in resistance to 6-Hz seizures, as compared to controls supplemented with vehicle solution (**Supplementary Fig. 8a, b**). Notably, we detected elevations in only serum acetate concentrations when SCFAs were supplemented in solid form in the paste diet (**Supplementary Fig. 8c,d**), which may be attributable to the rapid absorption, utilization, and distribution of exogenously delivered SCFAs^{45–47}. Consistent with this, we observed no overt differences in SCFA levels in fiber-supplemented mice fed KD (**Supplementary Fig. 9**), which may be due to variations in the timing of food intake and fiber metabolism relative to sample collection⁴⁵.“

We also added methodological descriptions for the LC-MS analysis of SCFA and samples in the Methods section in lines 660-676:

Short chain fatty acid (SCFA) measurements

The LC-MS analysis of SCFA in cecum and serum was performed by the Analytical Phytochemical Core at UCLA with modifications based on previously published protocols^{80,81}. Cecal contents were collected post-behavioral testing from terminal mouse dissections, immediately snap frozen in liquid nitrogen and stored at -80°C until analysis. Blood was collected via retro-orbital sampling as described above. 50 mg of cecal contents were homogenized in 200 μL PBS (Bead Mill, Fisher Scientific), vortexed and sonicated in ice water for 10min. The lysate was then centrifuged at $14,000 \times g$ for 10min at 4°C and the supernatant was collected. 30 μL serum or cecal lysate supernatant were mixed well with 30 μL isopropyl alcohol with isotope labeled internal standards (Acetic- d_3 , Propionic- d_5 and Butyric- d_7 $5\mu\text{g}/\text{mL}$ each), precipitated at -20°C overnight, and centrifuged at $14,000 \times g$ for 10min at 4°C . The supernatant was derivatized prior to LC-MS following previously published instructions^{80,81}. The LC-MS analysis was performed on an Agilent Zorbax SB-C18 2.1X150 mm column using the TSQ Quantum (Thermo-Finnigan) LC-ESI-MS/MS system at negative mode. Quantification was achieved by using Xcalibur data system. Calibration curves were built by fitting the analyte concentrations versus the peak area ratios of the analyte to isotope labeled internal standards. The peak area ratios target analyte to isotope-labeled internal standards in the samples were used to calculate the concentrations.

4) Additional metabolomics analysis to compare the metabolomes from gut samples from animals on the different diets would shed further light on which metabolites were enhanced or reduced and would add value to this study.

We thank the Reviewer for the suggestion. We agree that determining how fiber supplementation alters other non-SCFA metabolites would add value to the study. Therefore, we have performed targeted polar metabolomic profiling of cecal samples from mice fed KD4:1 + fiber mix vs KD4:1, following the methods now added to the Methods section lines 678-693:

Targeted polar metabolomic profiling

Polar metabolites were extracted based on previous publications with modifications⁸²⁻⁸⁴. Briefly, 50 mg of frozen cecal content was added to a pre-cooled bead beading tube containing 100 mg of 0.1 mm glass beads (Qiagen), 100 mg of 212-300 μm acid washed glass beads (Sigma-Aldrich), and 2 of 4 mm acid washed silica beads (OPS Diagnostics). 1000 μL of the extract solution (acetonitrile (ThermoScientific Chemicals, HPLC Grade)-methanol (Sigma-Aldrich, HPLC grade)-water (Fisher Chemical, HPLC Grade) = 2:2:1) was added into the tubes and homogenized in by bead beating (BioSpec Products) for several rounds of 30 sec at max speed with 5 min incubations on -20°C in between rounds until no particles were observed. Homogenate was incubated at -20°C overnight, then centrifuged at $16,000 \times g$ for 15 min at 4°C and stored at -20°C until analysis. The polar metabolite separation was done on HILIC column using a UPLC system equipped with mass spectrometer (Thermo Scientific) at the UCLA Metabolomics Center as previously described^{85,86}. Peaks were aligned among all samples and assigned identities using exact mass and retention time based on core's in-house database. Peaks were quantified by area under the curve integration and metabolites were filtered if less than 50% of the replicates have measurements.

Supplementary Figure 10. Select TCA metabolites and nucleotide derivatives are altered in cecum of mice fed with fiber mix supplemented KD4:1

a. Heatmap of polar metabolites belongs to 15 fiber-induced pathways identified in cecum of SPF mice fed KD4:1 and KD4:1+fiber mix as paste diet. Row names were colored according to superclass color scheme in Fig. 4d. Color scheme represents the z-scores based on the area under the curve across all samples within same row. Asterisk denotes statistical significance based on Wilcoxon signed-rank test, $p < 0.05$.

b. Amount (area under the curve) of select cecal metabolites in mice fed KD4:1 and KD4:1+fiber mix as paste diet. (n=4 cages/group, Wilcoxon signed-rank test. * $p < 0.05$. Data are presented as box-and-whisker plots with median and first and third quartiles)

The results, now provided as **Supplementary Fig. 10**, reveal that supplementation of KD4:1 with fiber mix leads to statistically significant decreases in cecal L-aspartate, hypoxanthine, xanthine, inosine, and uridine, and increases in 2-hydroxyglutarate. We added the following text describing these Results

Lines 310-316 Results:

“To explore additional non-SCFA metabolites that may mediate effects of fiber mix, we profiled an additional 93 cecal metabolites from mice fed KD4:1, with or without fiber

supplementation (**Supplementary Table S10**). Mice fed fiber-supplemented KD4:1 had significantly increased cecal levels of 2-hydroxyglutarate and decreased levels of L-aspartate, hypoxanthine, xanthine, inosine, and uridine (**Supplementary Fig. 10b**), which aligns with metagenomic alterations in genes related to TCA cycle and L-aspartate biosynthesis (**Fig. 3c**). These results additionally implicate alterations in adenosine and uracil metabolism.”

Lines 456-480 Discussion:

iv) In bacteria, the TCA pathway fuels aerobic respiration, wherein acetyl-CoA is converted to intermediate organic acids such as citrate, 2-oxoglutarate, and succinate. 2-oxoglutarate also known as α -ketoglutarate is an important intermediate in the TCA cycle and is known to regulate glutamate and GABA levels in the brain⁶². Cecal α -ketoglutarate levels did not differ in mice consuming KD4:1 supplemented with fiber mix, but its reduced product 2-hydroxyglutarate was significantly elevated (**Supplementary Fig. 10b**). How microbial levels of 2-oxoglutarate and other intermediate metabolites from the TCA cycle, may affect the brain to alter anti-seizure susceptibility is unknown. During KD therapy, patients supplemented with oral citrate as an alkalizing agent prevented metabolic acidosis without affecting the 7-month efficacy rates⁶³. Similarly, KD has been shown to affect the succinate levels through succinate dehydrogenase activity in rodent models of aging⁶⁴ and through effects on mitochondrial respiration to restore ATP production⁶⁵. v) Peripheral glutamine is imported by the brain and contributes to central metabolism of glutamate versus GABA, which has been implicated in the anticonvulsant effects of KD^{12,66}. Microbial genes related to glutamate and glutamine biosynthesis were positively associated with the presence of dietary fibers in clinical KDs and negatively associated with fiber-deficient seizure susceptible group KD3:1 (**Supplementary Fig. 2d,e**). We also observed reduced cecal L-aspartate levels with fiber mix supplementation (**Supplementary Fig. 10b**), which could be related to reduced glutamate transamination to aspartate⁶⁶. High levels of L-aspartate were reported in the cerebral cortex of epileptic individuals⁶⁷. Decreased brain aspartate levels were observed in mice fed KD for 3 days⁶⁸. Overall, these results suggest that increases in microbial biosynthetic pathways for preQ₀, L-alanine, and glutamate and glutamine biosynthesis, and reductions in microbial carbohydrate metabolism, may contribute to diet-induced seizure resistance. Further research is needed to determine whether there are causal links between these particular microbial functions and seizure protection.

5) The addition of SCFAs directly to the KD4:1 infant formula did not yield the same results as feeding the dietary fiber. What levels of SCFA were reached in the colon of fiber fed and SCFA fed mice? I did not see these data reported in the manuscript. Was it confirmed that the directly fed SCFA reached similar levels in the colon as were achieved by feeding dietary fiber? If not, the conclusion that the anti-seizure effects of the fiber containing KD4:1 were not mediated through SCFA was not fully demonstrated.

We appreciate the Reviewer’s concern about whether SCFA supplementation led to increased SCFA concentrations in the host. To address this, we measured serum SCFAs in mice exposed to either SCFAs supplemented in the drinking water and or in KD4:1 diet in solid form. We

observed that only acetate reached significantly higher concentrations compared to vehicle NaCl control when administered as a solid form in the diet. This is not particularly surprising to us as our prior studies and others' have suggested that SCFAs are rapidly absorbed and metabolized following exogenous supplementation and therefore their detection in the host will depend on the timing of observation relative to SCFA exposure. In our case, we collected samples from mice in the afternoon. Given that they were maintained on standard light cycle, this timing could reflect 6+ hours after their active period of feeding and drinking.

We added the following text in the Results section lines 304-310 and we modified Supplementary Figure 8:

“Notably, we detected elevations in only serum acetate concentrations when SCFAs were supplemented in solid form in the paste diet (**Supplementary Fig. 9d**), which may be attributable to the rapid absorption, utilization, and distribution of exogenously delivered SCFAs⁴⁵⁻⁴⁷.”

Supplementary Figure 8. SCFA supplementation does not phenocopy effects of fiber supplementation on KD-induced response to 6-Hz seizures.

a. 6-Hz seizure threshold (left) and latency to exploration (middle) for mice fed KD4:1 paste diet and supplemented with SCFAs or vehicle (NaCl) control in the drinking water (left, Welch's t-test $n=16$ mice/group, **** $p<0.0001$). Yellow line at $y = 10$ s represents threshold for scoring seizures. Average consumption of paste diets (right, $n=4$ cages; Wilcoxon signed-rank test). Data are presented as box-and-whisker plots with median and first and third quartiles.

b. 6-Hz seizure threshold (left) and latency to exploration (middle) for mice fed KD4:1 + SCFAs or vehicle (NaCl) control as a paste diet (left, Welch's t-test $n=16$ mice/group, *** $p<0.001$). Yellow line at $y = 10$ s represents threshold for scoring seizures. Average consumption of paste diets (right, $n=4$ cages; Wilcoxon signed-rank test). Data are presented as box-and-whisker plots with median and first and third quartiles).

c. SCFA concentration in serum and total SCFA concentration normalized to diet consumption per cage after 7 days of SCFAs or vehicle (NaCl) supplementation in drinking water ($n=4$

cages/group, Wilcoxon signed-rank test. Data are presented as box-and-whisker plots with median and first and third quartiles).

d. SCFA concentration in serum and total SCFA concentration normalized to diet consumption per cage after 7 days of SCFAs or vehicle (NaCl) supplementation in the paste diet (n=4 cages/group, Wilcoxon signed-rank test. Data are presented as box-and-whisker plots with median and first and third quartiles).

6) The model human infant microbial community included the following strains, based on their high relative abundance in their respective phyla and prevalence at >1% relative abundance across the study population.

Type strains from either from ATCC or DSMZ collection
Bifidobacterium longum subsp. infantis DSM 20088,
Bifidobacterium longum subsp. longum ATCC BAA-999,
Bifidobacterium breve DSM 20213,
Bacteroides fragilis ATCC 25285,
Bacteroides vulgatus ATCC 8482,
Enterococcus faecalis ATCC 19433,
Clostridium perfringens ATCC 13124,
Escherichia coli K-12 ATCC 10798,
Klebsiella pneumoniae subsp. pneumoniae ATCC 13883.

Were they each tested against the other, to ensure that no strain produced antimicrobial activity against others in the community?

We thank the Reviewer for raising this important point. Prior to initiating experiments related to the KD, we tested the species against each other by culturing them in a rich media that would support the growth of all species. The composition of rich media is given Supplementary Data 2. We confirmed in Supplementary Figure 3b that none of the species' growth was suppressed after 24 h culture in the rich medium. We calculated the absolute concentrations from qPCR quantification when grown in rich media after subtracting the pre-fermentation concentrations.